# Investigating Variance Definitions for Stochastic Mirror Descent with Relative Smoothness

## Abstract

Mirror Descent is a popular algorithm, that extends Gradients Descent (GD) beyond the Euclidean geometry. One of its benefits is to enable strong convergence guarantees through smooth-like analyses, even for objectives with exploding or vanishing curvature. This is achieved through the introduction of the notion of *relative smoothness*, which holds in many of the common use-cases of Mirror descent. While basic deterministic results extend well to the relative setting, most existing stochastic analyses require additional assumptions on the mirror, such as strong convexity (in the usual sense), to ensure bounded variance. In this work, we revisit Stochastic Mirror Descent (SMD) proofs in the (relatively-strongly-) convex and relatively-smooth setting, and introduce a new (less restrictive) definition of variance which can generally be bounded (globally) under mild regularity assumptions. We then investigate this notion in more details, and show that it naturally leads to strong convergence guarantees for stochastic mirror descent. Finally, we leverage this new analysis to obtain convergence guarantees for the Maximum Likelihood Estimator of a Gaussian with unknown mean and variance.

## 1 Introduction

The central problem of this paper is to solve optimization problems of the following form:

$$\min_{x \in C} f(x), \text{ where } f(x) = \mathbb{E}\left[f_\xi(x)\right], \tag{1}$$

where $C$ is a closed convex subset of $\mathbb{R}^d$, and $f_\xi$ are differentiable convex functions (stochasticity is on the variable $\xi$). The problems that we will consider typically arise from machine-learning use-cases, meaning that the dimension $d$ can be very large. Therefore, first-order methods are popular for solving these problems, since they usually scale well with the dimension.

In standard machine learning setups, computing a gradient of $f$ is very costly (or even impossible), since it requires computing gradients for all individual examples in the dataset. Yet, gradients of $f_\xi$ are relatively cheap, and arbitrarily high precisions are generally not required. This makes Stochastic Gradient Descent (SGD) the method of choice [4]. Using a step-size $\eta > 0$, the SGD update from point $x \in \mathbb{R}^d$ can be written as $x_{\text{SGD}}^+ = \arg\min_{u \in C} \left\{ \eta \nabla f_\xi(x)^\top u + \frac{1}{2}\|u - x\|^2 \right\}$.

While the standard Euclidean geometry leading to Gradient Descent (GD) fits many use-cases quite well, several applications are better solved with *Mirror Descent* (MD), a generalization of GD which allows to better capture the geometry of the problem. For instance, the Kullback-Leibler divergence might be better suited to discriminating between probability distributions than the (squared) Euclidean norm, and this is something that one can leverage using MD with entropy as a mirror. As a matter of fact, many standard algorithms can be interpreted as MD, *i.e.*, as generalized first-order methods. This is for instance the case in statistics, where Expectation Minimization and Maximum A Posteriori

estimators can be interpreted as running MD with specific mirror and step-sizes [15, 17]. Mirror descent can also be used to solve Poisson inverse problems, which have many applications in astronomy and medicine [3], to reduce the communication cost of distributed algorithms [24, 12], or to solve convex quartic problems [6]. In the online learning community as well, many standard algorithms such as Exponential Weight Updates or Follow-The-Regularized-Leader can be interpreted as running mirror descent [21, 13]. There are still many open questions regarding the convergence guarantees for most of the algorithms mentioned above. Therefore, progress on the understanding of MD can lead to a plethora of results on these applications, and more generally to a more consistent theory for Majorization-Minimization algorithms. This paper is a stepping stone in this direction.

Let us now introduce the *mirror map*, or *potential* function $h$, together with the *Bregman divergence* with respect to $h$, which is defined for $x, y \in \operatorname{dom} h$ as $D_h(x, y) = h(x) - h(y) - \nabla h(y)^\top (x - y)$. We now introduce the Stochastic Mirror Descent (SMD) update, which can be found in its deterministic form in, *e.g.*, Nemirovskij and Yudin [22]. SMD consists in replacing the squared Euclidean norm from the SGD update by the Bregman divergence with respect to the mirror map $h$:

$$x^+(\eta, \xi) = \arg \min_{u \in C} \left\{ \eta \nabla f_\xi(x)^\top u + D_h(u, x) \right\}. \tag{2}$$

Note that since $D_{\|\cdot\|^2}(x, y) = \|x - y\|^2$, one can recover SGD by taking $h = \frac{1}{2} \| \cdot \|^2$. In this sense, SMD can be viewed as standard SGD, but changing the way distances are computed, and so the geometry of the problem. Yet, this change significantly complicates the convergence analysis of the method, since the Bregman divergence, *in general*: *(i)* does not satisfy the triangular inequality, *(ii)* is not symmetric, *(iii)* is not translation-invariant, *(iv)* is not convex in its second argument.

This means that analyzing mirror descent methods requires quite some care, and that many standard (S)GD results do not extend to the mirror setting. For instance, one can prove that mirror descent cannot be *accelerated* in general [8]. Similarly, applying techniques such as variance-reduction requires additional assumptions [7]. To ensure that $x^+(\eta, \xi)$ exists and is unique, we first make the following blanket assumption throughout the paper:

**Assumption 1.** *Function $h : \mathbb{R}^d \to \mathbb{R} \cup \{\infty\}$ is twice continuously differentiable and strictly convex on $C$. For every $y \in \mathbb{R}^d$, the problem $\min_{x \in C} h(x) - x^\top y$ has a unique solution, which lies in $\operatorname{int} C$, and all $f_\xi$ are convex.*

Note that the regularity assumption on $h$ could be relaxed, as discussed in Section 3, but we choose a rather strong one to make sure all the objects we will manipulate are well-defined. Interestingly, while mirror descent changes the way distances are computed to move away from the Euclidean geometry, standard analyses of mirror descent methods, and in particular in the online learning community, still require strong convexity and Lipschitz continuity with respect to norms [5, Chapter 4]. It is only recently that a *relative smoothness* assumption was introduced to study mirror descent [2, 20], together with the corresponding relative strong convexity.

**Definition 1.** *The function $f$ is said to be $L$-relatively smooth and $\mu$-relatively strongly convex with respect to $h$ if for all $x, y \in C$: $\mu D_h(x, y) \le D_f(x, y) \le L D_h(x, y)$. To lighten notation, we will omit the dependence on $h$ and simply write that $f$ is $L$-rel.-smooth unless clearly specified.*

Definition 1 extends the standard smooth and strongly convex assumptions that correspond to the case $h = \frac{1}{2} \| \cdot \|^2$, so that for all $x \in C$, $\nabla^2 h(x) = I$ the identity matrix. These assumptions allow MD analyses to generalize standard GD analyses, and in particular to obtain similar linear and sublinear rates, with constant step-size and conditions adapted to the *relative* assumptions.

While the basic deterministic setting is now well-understood under relative assumptions, a good understanding of the stochastic setting remains elusive. In particular, as we will see in more details in the related work section, *all existing proofs somehow require the mirror $h$ to be globally strongly convex with respect to a norm*, or have non-vanishing variance. The only case that can be analyzed tightly is under *interpolation* (there exists a point that minimizes all stochastic functions), or when using Coordinate Descent instead of SMD [10, 11]. This is a major weakness, as the goal of relative smoothness is precisely to avoid comparisons to norms. Indeed, even when these "absolute" regularity assumptions hold, the smoothness and strong convexity constants are typically very loose, and the theory is not representative of the observed behaviour of the algorithms.

However, as hinted at earlier, this was expected: acceleration is notoriously hard to achieve for mirror descent (and even impossible in general [8]), and variance reduction typically encounters the same

problems [7]. For stochastic updates, this comes from the fact that it is impossible to disentangle the stochastic gradient from the effect of the curvature of $h$ at the point at which it is applied.

**Contribution and outline.** The main contribution of this paper is to introduce a new analysis for mirror descent, with a variance notion which is provably bounded under mild regularity assumptions: typically, the same as those required for the deterministic case. We introduce our new variance notion, and compare it with standard ones from the literature in Section 2. This new analysis is both simpler and tighter than existing ones, as shown in Section 3. Finally, we use our results to analyse the convergence of the Maximum Likelihood and Maximum A Posteriori estimators for a Gaussian with unknown mean and variance in Section 4, and show that it is the first generic stochastic mirror descent analysis that obtains meaningful finite-time convergence guarantees in this case.

# 2 Variance Assumptions

We now focus on the various variance assumptions under which Stochastic Mirror Descent is analyzed. Some manipulations require technical lemmas, such as the duality property of the Bregman divergence or the Bregman co-coercivity lemma, which can be found in Appendix A.

We start by introducing our variance definition, prove a few good properties for it, and then compare it with the existing ones to highlight their shortcomings. The two key properties we would like to ensure (and which are not satisfied by other definitions) are: (i) boundedness without strong convexity of $h$ or restricting the SMD iterates, and (ii) finiteness for $\eta \to 0$ (with the appropriate scaling).

## 2.1 New variance definition

Let $\eta > 0$, and recall that $x^+(\eta, \xi)$ is the result of a SMD step from $x$ using function $f_\xi$ with step-size $\eta$ (Equation (2)). From now on, when clear from the context, we will simply denote this point $x^+$. Yet, although the dependence is now implicit, do keep in mind that $x^+$ is a stochastic quantity that is not independent from $\xi$ nor $\eta$, as this is critical in most results. Under Assumption 1, $x^+$ writes:

$$\nabla h(x^+) = \nabla h(x) - \eta \nabla f_\xi(x). \tag{3}$$

Similarly, we denote by $\overline{x^+}$ the deterministic Mirror Descent update, which is such that $\nabla h(\overline{x^+}) = \nabla h(x) - \eta \nabla f(x)$. We also introduce $h^* : y \mapsto \arg\max_{x \in C} x^\top y - h(x)$ the convex conjugate of $h$, which verifies $\nabla h^*(\nabla h(x)) = x$. Let us now define the key function

$$f_\eta(x) = f(x) - \frac{1}{\eta} \mathbb{E}\left[D_h(x, x^+)\right]. \tag{4}$$

**Definition 2.** We define the variance of the stochastic mirror descent iterates given by (2) as $\sigma_{\star,\eta}^2 = \frac{1}{\eta} \sup_{x \in C} (f(x_\star) - f_\eta(x)) = \frac{f^\star - f_\eta^\star}{\eta}$, where $f^\star$ and $f_\eta^\star$ are respectively the inf. of $f$ and $f_\eta$.

We now state various bounds on $\sigma_{\star,\eta}^2$, to help understand its behaviour. We start by positivity, which is an essential property that justifies the square in the definition.

**Proposition 2.1** (Positivity). *For all $\eta > 0$, $\sigma_{\star,\eta} \geq 0$.*

This result follows from $f_\eta(x) \leq f(x)$, since $D_h(x, x^+) \geq 0$ for all $x \in C$ by convexity of $h$.

**Stochastic functions after a step.** We first upper bound $\sigma_{\star,\eta}^2$ directly in terms of $f_\xi$.

**Proposition 2.2.** *If $f_\xi$ is $L$-rel.-smooth and $\eta \leq 1/L$, then $\sigma_{\star,\eta}^2 \leq \frac{1}{\eta}\left(f(x_\star) - \min_{x \in C} \mathbb{E}\left[f_\xi(x^+)\right]\right)$.*

*Proof.* Since $D_h(x, x^+) = \langle \nabla h(x^+) - \nabla h(x), x^+ - x \rangle - D_h(x^+, x)$, then $D_h(x, x^+) = -\eta \nabla f_\xi(x)^\top (x^+ - x) - D_h(x^+, x) = \eta \left(D_{f_\xi}(x^+, x) - f_\xi(x^+) + f_\xi(x)\right) - D_h(x^+, x)$. The relative smoothness of $f_\xi$ and the step-size condition imply that $\eta D_{f_\xi}(x^+, x) \leq D_h(x^+, x)$, leading to $\frac{1}{\eta} D_h(x, x^+) \leq f_\xi(x) - f_\xi(x^+)$, and the result follows. $\square$

This bound offers a new point of view on the variance, which can be bounded as the difference between the optimum of $f$, and the optimum of a related function, in which we make one mirror descent step before evaluating each $f_\xi$.

**Finiteness.** Proposition 2.2 implies the following:

**Corollary 2.3.** *If $f_\xi$ is L-relatively-smooth w.r.t. $h$ and admits a minimum $x_\star^\xi \in \text{int } C$ a.s., then for all $\eta \leq 1/L$, $\sigma_{\star,\eta}^2 \leq \frac{f(x_\star) - \mathbb{E}\left[f_\xi(x_\star^\xi)\right]}{\eta}$. In particular, $\sigma_{\star,\eta}^2$ is finite.*

This result directly comes from the fact that $\min_{x \in C} \mathbb{E}\left[f_\xi(x^+)\right] \geq \mathbb{E}\left[\min_{x \in C} f_\xi(x^+)\right] \geq \mathbb{E}\left[f_\xi(x_\star^\xi)\right]$. It shows that the standard regularity assumptions for the convergence of stochastic mirror descent guarantee that *the variance as introduced in Definition 2 remains bounded.* This is a strong result, that justifies the supremum in the variance definition. Indeed, **most other variance definitions require additional assumptions for the variance to remain bounded after the supremum**. Instead, we *globalize* the variance definition, by taking the supremum over the right quantity to ensure that it remains bounded over the whole domain without having to explicitly assume it.

Note that the bound from Corollary 2.3 has already been investigating in other settings for stochastic optimization [19], as discussed in Section 2.2. While useful to show boundedness, this bound has a major drawback, which is that it explodes when the step-size $\eta$ vanishes. This does not reflect what happens in practice, which is why we investigate finer bounds on $\sigma_{\star,\eta}^2$.

**Gradient norm at optimum.**   A usual way of formulating variance is to express it as the norm of the difference between stochastic gradients and the deterministic gradients. While the previous bounds highlight dependencies on the gradient steps (through evaluations at $x^+$), none of them really corresponds to "the size of the stochastic gradients at optimum". The key subtlety is that when using mirror descent, it is important to also specify the point at which these gradients are applied, and the following proposition gives a bound of this flavor on $\sigma_{\star,\eta}^2$. In this section, $x_\eta$ denotes the minimizer of $f_\eta$ when it exists and is in $\text{int } C$. Otherwise, unless explicitly stated, results involving $x_\eta$ can be replaced by a limit for $x \to x_\eta$.

**Proposition 2.4.** *If $f$ is L-rel.-smooth, $\eta \leq 1/L$ and $x_\star \in \text{int } C$, $\sigma_{\star,\eta}^2 \leq \frac{1}{\eta^2}\mathbb{E}\left[D_h\left(\overline{x_\eta^+}, x_\eta^+\right)\right]$.*

This can be considered as the Mirror Descent equivalent of $\mathbb{E}\left[\|\nabla f_\xi(x_\star)\|^2\right]$. Yet, a key difference is that stochastic gradients are evaluated at point $x_\eta$ instead of $x_\star$, and $\nabla f(x_\eta) \neq 0$ in general.

*Proof.* For all $x$, applying the duality property of the Bregman divergence leads to:

$$\mathbb{E}\left[D_h(x, x^+)\right] = \mathbb{E}\left[D_{h^*}(\nabla h(x^+), \nabla h(x))\right] = \mathbb{E}\left[D_{h^*}(\nabla h(x) - \eta\nabla f_\xi(x), \nabla h(x))\right]$$
$$= \mathbb{E}\left[D_{h^*}(\nabla h(x) - \eta\nabla f(x), \nabla h(x))\right] + \mathbb{E}\left[D_{h^*}(\nabla h(x) - \eta\nabla f_\xi(x), \nabla h(x) - \eta\nabla f(x))\right]$$
$$= \mathbb{E}\left[D_{h^*}(\nabla h(x) - \eta\left[\nabla f(x) - \nabla f(x_\star)\right], \nabla h(x))\right] + \mathbb{E}\left[D_h\left(\overline{x^+}, x^+\right)\right],$$

where the last equality comes from the Bregman bias-variance decomposition Lemma [23]. We then use the Bregman cocoercivity Lemma [7] to obtain: $\mathbb{E}\left[D_h(x, x^+)\right] \leq \eta D_f(x, x_\star) + \mathbb{E}\left[D_h\left(\overline{x^+}, x^+\right)\right]$. All these technical results can be found in Appendix A. In the end, $f_\eta(x) \geq f(x_\star) - \frac{1}{\eta}\mathbb{E}\left[D_h(\overline{x^+}, x^+)\right]$, and this is in particular true for $x = x_\eta$. $\qquad\square$

**Limit behaviour.** A first observation is that both the $D_h(x, x^+)$ term in the definition of $f_\eta$ and our variance definition are scaled by $\eta^{-1}$. Yet, they remain finite when $\eta \to 0$. While this is clear in the Euclidean setting, this property holds more generally, as shown in the two following results.

**Proposition 2.5.** *Let $x \in C$ and $\eta_0 > 0$ s.t. $\mathbb{E}D_h(x, x^+(\eta_0, \xi)) < \infty$. Then, $f_\eta(x) \xrightarrow{\eta \to 0} f(x)$.*

Note that uniform convergence of $f_\eta$ to $f$ would require that there exists $\eta > 0$ such that $\sup_{x \in C} D_h(x, x^+)$ is finite, which we cannot guarantee in general (it does not hold for $f = g = \frac{1}{2}\|\cdot\|^2$ defined on $\mathbb{R}^d$ for instance). Denote $\|x\|_A^2 = x^\top A x$, then:

**Proposition 2.6** (Small step-sizes limit). *If $f_\xi$ are L-rel.-smooth and $f$ has a unique minimizer $x_\star$ and for some $\eta_0 > 0$, $x_\eta = \arg\min f_\eta(x)$ exists and is in $\text{int } C$ for $\eta \leq \eta_0$,*

$$\lim_{\eta \to 0} \sigma_{\star,\eta}^2 = \lim_{\eta \to 0} \frac{1}{\eta^2}\mathbb{E}\left[D_h(x_\star^+, x_\star)\right] = \frac{1}{2}\mathbb{E}\left[\|\nabla f_\xi(x_\star)\|_{\nabla^2 h(x_\star)^{-1}}^2\right]. \tag{5}$$

This variance is actually the best we can hope for in the Bregman setting, which indicates the relevance of Definition 2. Indeed, this term exactly correspond to the variance one would obtain when making infinitesimal SMD steps from $x_\star$, *i.e.*, the norm of the stochastic gradients at optimum in the geometry given by $\nabla^2 h(x_\star)^{-1}$.

## 2.2 Standard Assumptions

We now compare Definition 2 with several variance assumptions from the literature. Note that they typically "only" require the bounds to hold for all iterates over the trajectory. However, in the absence of proof that the iterates stay in certain regions of the space, suprema over the whole domain are required for all variance definitions.

**Euclidean case.** Let us now take a step back and look at the Euclidean case, $h = \frac{1}{2}\|\cdot\|^2$, and assume that $f$ is $L$-smooth. Writing Equation (3) with this specific $h$ and replacing $x_\eta$ by a supremum, we obtain $\sigma_{\star,\eta}^2 \leq \sup_{x \in C} \mathbb{E}\left[\frac{1}{2}\|\nabla f(x) - \nabla f_\xi(x)\|^2\right]$, which is a common though debatable variance assumption. Indeed, it involves a maximum over the domain, and is in particular not bounded in general even for simple examples like Linear Regression. Yet, we can recover another standard variance assumption by assuming the smoothness of all $f_\xi$ [9], which writes $\sigma_{\star,\eta}^2 \leq \mathbb{E}\left[\|\nabla f_\xi(x_\star)\|^2\right]$.

This result is obtained by writing that $\|\nabla f_\xi(x)\|^2 \leq 2\|\nabla f_\xi(x) - \nabla f_\xi(x_\star)\|^2 + 2\|\nabla f_\xi(x_\star)\|^2$, and bounding the first term using smoothness. In particular, we see that standard Euclidean variance definitions are natural bounds of $\sigma_{\star,\eta}^2$. Detailed derivations can be found in Appendix B.

**Divergence between stochastic and deterministic gradients.** An early variance definition for SMD in the relative setting comes from Hanzely and Richtárik [10], who define $\sigma_{\text{sym}}^2$ as:

$$\sigma_{\text{sym}}^2 = \frac{1}{\eta} \sup_{x \in C} \mathbb{E}\left[\left\langle \nabla f(x) - \nabla f_\xi(x), x^+ - \overline{x^+} \right\rangle\right] = \frac{1}{\eta^2} \sup_{x \in C} \mathbb{E}\left[D_h\left(x^+, \overline{x^+}\right) + D_h\left(\overline{x^+}, x^+\right)\right],$$

where we recall that $\overline{x^+}$ is such that $\nabla h(\overline{x^+}) = \nabla h(x) - \eta \nabla f(x)$. We remark two main things when comparing $\sigma_{\text{sym}}^2$ with Proposition 2.4: (i) $\sigma_{\star,\eta}^2$ is not symmetrized, and contains only one of the two terms, and (ii) the bound only needs to hold at $x_\eta$ instead of for all $x \in C$. As a result, we directly obtain that $\sigma_{\star,\eta}^2 \leq \sigma_{\text{sym}}^2$, and $\sigma_{\text{sym}}^2$ is actually infinite in most cases, whereas $\sigma_{\star,\eta}^2$ is usually finite, as seen above.

**Stochastic gradients at optimum.** Dragomir et al. [7] define the variance as:

$$\sigma_{DEH}^2 = \sup_{x \in C} \frac{1}{2\eta^2} \mathbb{E}\left[D_{h^*}(\nabla h(x) - 2\eta \nabla f_\xi(x_\star), \nabla h(x))\right] = \sup_{x \in C} \mathbb{E}\left[\|\nabla f_\xi(x_\star)\|_{\nabla^2 h^*(z(x))}^2\right],$$

where $z(x) \in [\nabla h(x), \nabla h(x) - \eta \nabla f_\xi(x_\star)]$ The main interest of this definition is that stochastic gradients are only taken at $x_\star$. In particular, this variance is 0 in case there is interpolation (all stochastic functions share a common minimum). However, this quantity can blow up if $h$ is not strongly convex, since in this case $\nabla^2 h^*$ is not upper bounded (indeed, smoothness of the conjugate is ensured by strong convexity of the primal function [14]). Following similar derivations, but *after the supremum has been taken*, we arrive at:

**Proposition 2.7.** *If $f$ is $L$-relatively-smooth w.r.t. $h$, then for $\eta < 1/(2L)$ and some $z_\eta \in [\nabla h(x_\eta), \nabla h(x_\eta) - \eta \nabla f_\xi(x_\star)]$, the variance can be bounded as $\sigma_{\star,\eta}^2 \leq \mathbb{E}\left[\|\nabla f_\xi(x_\star)\|_{\nabla^2 h^*(z_\eta)}^2\right]$.*

In particular, we obtain a finite bound without having to restrict the space.

**Functions variance.** Another variance definition that appears in the SGD literature is of the form $f(x_\star) - \mathbb{E}\left[f_\xi(x_\star^\xi)\right]$, using the optima of the stochastic functions [19]. Unfortunately, the results derived with this definition do not obtain a vanishing variance term when $\eta \to 0$, unlike most other variance definitions, and contrary to what is observed in practice, that smaller step-sizes reduce the variance. The vanishing variance term can be obtained by rescaling by $1/\eta$ (so considering $\left(f(x_\star) - \mathbb{E}\left[f_\xi(x_\star^\xi)\right]\right)/\eta$ instead), but this variance definition would explode for $\eta \to 0$. This is because using such a definition would come down to performing the supremum step within the expectation from Proposition 2.2, using that $f_\xi(x^+) \geq f_\xi(x_\star^\xi)$, which is a very crude bound. Instead, Corolary 2.3 directly shows that our variance definition is tighter than this one, and in particular (i) it is bounded for all $\eta > 0$, (ii) it remains finite as $\eta \to 0$ even with the proper rescaling (Proposition 2.6).

**Relation to $c$-transform.** Mirror descent can be viewed as an alternate minimization method on transforms of $f$ [18]. This point of view subsumes many methods, including the Newton Method or Mirror Descent. Central to their analysis is the notion of $c$-transform $f^c(y) = \sup_{x \in C} f(x) - c(x, y)$, a standard quantity from optimal transport [25]. It turns out that for $\eta \leq 1/L$, $f_\eta$ is actually linked to the $c$-transform as $f_\eta(x) = \mathbb{E}\left[ f_\xi^c(x^+) \right]$, where we use the cost $c(x, y) = \frac{1}{\eta} D_h(x, y)$. Since $f(x_\star) = f^c(x_\star) = \arg\min_{x \in C} f(\overline{x^+})$, denoting $\mathcal{T}_c(g) = g^c(\nabla h^*(\nabla h(x) - \eta \nabla g(x)))$, we have that $\sigma_{\star, \eta}^2 = \frac{1}{\eta}(\min_{x \in C} \mathcal{T}_c(\mathbb{E}\left[ f_\xi \right])(x) - \min_{x \in C} \mathbb{E}\left[ \mathcal{T}_c(f_\xi) \right](x))$. We recognize the structure of a variance, as the difference between an operator applied to the expectation of a random variable, and the expectation of the operator applied to the random variable. Yet, compared to standard (Euclidean) analyses of SGD, it does not simply corresponds to the variance of the stochastic gradients (at optimum), and bears a more complex form.

In this section, we have highlighted the connections with other definitions, and argued that $f_\eta$ (and its minimum) is a relevant quantity. In particular, Definition 2 is the only definition that allows boundedness of the variance notion both after a supremum step over the iterates (and without strong convexity of $h$) and in the $\eta \to 0$ limit with the proper rescaling.

# 3    Convergence Analysis

Now that we have (extensively) investigated $\sigma_{\star, \eta}^2$, and the various interpretations that come from different bounds, we are ready to state the convergence results. Some proofs in this section are just sketched, but complete derivations can be found in Appendix C.

## 3.1    Relatively Strongly Convex setting.

Recall that $f_\eta^\star = \inf_{x \in C} f_\eta(x)$. Starting from an arbitrary $x^{(0)}$, the sequence $(x^{(k)})_{k \geq 0}$ is built as $x^{(k+1)} = (x^{(k)})^+$ for $k \in \{0, T\}$ for some $T > 0$

**Theorem 3.1.** *If $f$ is $\mu$-relatively-strongly-convex with respect to $h$, under a constant step-size $\eta$, the iterates obtained by SMD (Equation (3)) verify*

$$\eta \left[ \mathbb{E}\left[ f_\eta(x^{(T)}) \right] - f_\eta^\star \right] + \mathbb{E}\left[ D_h(x_\star, x^{(T+1)}) \right] \leq (1 - \eta\mu)^{T+1} D_h(x_\star, x^{(0)}) + \frac{\eta \sigma_{\star, \eta}^2}{\mu}. \quad (6)$$

Note that the (relatively) strongly-convex theorem has a standard form, and recovers usual MD results if we remove the variance, and standard SGD results if we take $h = \frac{1}{2} \| \cdot \|^2$.

*Proof of Theorem 3.1.* We start from a variation of Dragomir et al. [7, Lemma 4]:

$$\mathbb{E}\left[ D_h(x_\star, x^+) \right] - D_h(x_\star, x) + \eta D_f(x_\star, x) = -\eta[f(x) - f(x_\star)] + \mathbb{E}\left[ D_h(x, x^+) \right] \quad (7)$$

$$= \eta \left[ f(x_\star) - \left( f(x) - \frac{1}{\eta} \mathbb{E}\left[ D_h(x, x^+) \right] \right) \right] = \eta \left[ f(x_\star) - f_\eta(x) \right] \quad (8)$$

$$= -\eta \left[ f_\eta(x) - f_\eta^\star \right] + \eta \left[ f(x_\star) - f_\eta^\star \right]. \quad (9)$$

Using that $D_f(x_\star, x) \geq \mu D_h(x_\star, x)$, and remarking that $f(x_\star) - f_\eta^\star = \eta \sigma_{\star, \eta}^2$, we obtain:

$$\eta \left[ f_\eta(x) - f_\eta^\star \right] + \mathbb{E}\left[ D_h(x_\star, x^+) \right] \leq (1 - \eta\mu) D_h(x_\star, x) + \eta^2 \sigma_{\star, \eta}^2. \quad (10)$$

At this point, we can neglect the $\eta \left[ f_\eta(x) - f_\eta^\star \right] \geq 0$ terms and chain the inequalities for $x = x^{(t)}$ for $t$ from 0 to $T$ to obtain the result. □

This proof is quite simple, and naturally follows from Lemma C.1. One can also note that *relative smoothness of $f$ is not required to obtain Theorem 3.1*, which has no condition on the step-size. This is not a typo, but reflects the fact that *step-size conditions are needed to obtain a bounded variance*. Indeed, the variance as defined here entangles aspects tied with the error due to discretization (which is usually dealt with using smoothness), and the error due to stochasticity. This is natural, as the stochastic noise vanishes in the continuous limit ($\eta \to 0$). Besides, the magnitude of the updates depends both on where the stochastic gradient is applied and on the step-size. Yet, the simplicity of

the proof is partly due to this entanglement, meaning that we have deferred some of the complexity to the bounding of the variance term.

Also note that Theorem 3.1 uses constant step-sizes, but Equation (10) can be used with time-varying step-sizes, as is done for instance in the proof of Theorem 4.3. A variant of Theorem 3.1 in which the discretization error is partly removed from the notion of variance writes:

**Corollary 3.2.** *Let $f$ be $\mu$-strongly-convex and $L$-relatively-smooth with respect to $h$, and $f_+^\star = \inf_{x \in C} \mathbb{E}\left[f_\xi(x^+)\right]$. If $\eta \leq 1/L$, the SMD iterates (Equation (3)) with constant step-size $\eta$ verify*

$$\eta\left[\mathbb{E}\left[f_\xi((x^{(T)})^+)\right] - f_+^\star\right] + \mathbb{E}\left[D_h(x_\star, x^{(T+1)})\right] \leq (1-\eta\mu)^{T+1}D_h(x_\star, x^{(0)}) + \frac{\eta}{\mu}\left[\frac{f(x_\star) - f_+^\star}{\eta}\right].$$

This alternate version is obtained using that $f_\eta(x) \geq \mathbb{E}\left[f_\xi(x^+)\right]$, a key step from the proof of Proposition 2.2 (see (8)). In the deterministic case, $f_+^\star = f(x_\star)$, and we recover standard results.

## 3.2 Convex setting.

Let us now consider the convex case, meaning that $\mu = 0$.

**Theorem 3.3.** *If $f$ is convex, the iterates obtained by SMD using a constant step-size $\eta > 0$ verify*

$$\frac{1}{T+1}\sum_{k=0}^{T}\mathbb{E}\left[f_\eta(x^{(k)}) - f_\eta^\star + D_f(x_\star, x^{(k)})\right] \leq \frac{D_h(x_\star, x^{(0)})}{\eta(T+1)} + \eta\sigma_{\star,\eta}^2. \tag{11}$$

This theorem is obtained by summing Equation (9) for $x = x^{(k)}$ for all $k \in \{1, \ldots, T\}$ and rearranging the terms. Note that varying step-size results can be obtained in the same way.

This case differs from standard convex analyses, in that we obtain a control on $f_\eta(x^{(k)}) - f_\eta^\star + D_f(x_\star, x^{(k)})$ instead of the usual $f(x^{(k)}) - f(x_\star)$. One of the main consequences is that we cannot get a control on the average iterate since Bregman divergences are in general not convex in their second argument, and $f_\eta$ is not necessarily convex. This non-standard result is a direct consequence of our choice of variance definition, but it is actually a quantity that naturally arises in the analysis. Note that a variant involving $f_+^\star$ can be obtained in the same lines as Corollary 3.2.

**Controlling $f_\eta$.** The results in this section do not directly control the function gap $f(x) - f^*$, but rather the transformed one $f_\eta(x) - f_\eta^\star$. Yet, the continuity result (in $\eta$) from Proposition 2.5 shows that the bounds we provide can still be interpreted as relevant function values for small $\eta$.

**Controlling $D_f(x_\star, x^{(k)})$.** An interesting property of $D_f(x_\star, x^{(k)})$ is that it can be linked with the size of the gradients of $f$, as shown by the following result.

**Proposition 3.4.** *If $\nabla f(x_\star) = 0$ and $f$ is $L$-relatively smooth with respect to $h$ then for all $x \neq x_\star$,* $D_f(x_\star, x) \geq LD_{h^*}\left(\nabla h(x_\star) + \frac{\nabla f(x)}{L}, \nabla h(x_\star)\right) > 0.$

This is a Bregman equivalent of controlling the gradient squared norm, with the additional benefit that the reference point at which we apply the gradient is the optimum $x_\star$. Besides, Proposition 3.4 shows that $D_f(x_\star, x) > 0$ for $x \neq x_\star$ without requiring $f$ to be strictly convex (only $h$).

**Minimal assumptions on $h$.** Note that the theorems in this section do not actually require $h$ to satisfy Assumption 1, but only that iterations can be written in the form of Equation 3 (which is guaranteed by Assumption 1). While Assumption 1 allows for instance to use the Bregman cocoercivity lemma with any points, or ensures that $\nabla^2 h$ is well-defined, which we leverage extensively in Section 2, our theorems are much more general than this, and include applications such as proximal gradient mirror descent (next remark) or the MAP for Gaussian Parameters Estimation (next section).

**Stochastic Mirror Descent with a Proximal term.** Note that our results can be directly extended to handle a proximal term (similarly to the Euclidean proximal gradient algorithm), to handle composite objectives of the form $f + g$ (and in particular projections, for cases in which $g$ is the indicator of a convex set). More details can be found in Appendix E.

## 4  MAP For Gaussian Parameters Estimation.

So far, we have proposed new variance definitions for the analysis of stochastic mirror descent, and we have shown that they compare favorably to existing ones, while leading to simple convergence proofs. In this section, we investigate the open problem formulated by Le Priol et al. [17], which is to find non-asymptotic convergence guarantees for the KL-divergence of the Maximum A Posteriori (MAP) estimator. In particular, this example highlights the relevance of the infimum step on $f_\eta$, since it gives the first generic analysis that obtains meaningful finite time convergence rates.

### 4.1  MAP and MLE of exponential families.

We now rapidly review the formalism of exponential families. More details can be found in Le Priol et al. [17], and Wainwright et al. [26, Chapter 3]. Let $X$ be a random variable, and $T$ a deterministic function, then the density of an exponential family for a sample $x$ writes $p_\theta(x) = p(x|\theta) = \exp(\langle \theta, T(x)\rangle - A(\theta))$, where $A$ is often refered to as the log-partition function. In this case, $\theta$ is called the natural parameter, and $T$ is the sufficient statistic. Function $A$ is convex, and we can thus establish a form of duality through convex conjugacy. The *entropy* writes $A^*(\mu) = \max_{\theta' \in \Theta} \langle \mu, \theta'\rangle - A(\theta')$. Parameter $\mu$ is called the *mean* parameter, and the standard MAP estimator can be derived for $n_0 \in \mathbb{N}$, $\mu_0 \in \mathbb{R}$ as $\mu_{\mathrm{MAP}}^{(n)} = \frac{n_0 \mu^{(0)} + \sum_{i=1}^n T(X_i)}{n_0 + n}$. The Maximum Likelihood Estimator (MLE) corresponds to taking $n_0 = 0$. An interesting observation is that $\mu_{\mathrm{MAP}}^{(n)}$ can be obtained recursively for $n > 0$, as $\mu_{\mathrm{MAP}}^{(0)} = \mu^{(0)}, \eta_n = (n+n_0)^{-1}, \mu_{\mathrm{MAP}}^{(n+1)} = \mu_{\mathrm{MAP}}^{(n)} - \eta_n \nabla g_{X_n}(\mu_{\mathrm{MAP}}^{(n)})$, with $\nabla g_{X_n}(\mu) = \mu - T(X_n)$. In terms of primal variable $\theta^{(n)} = \nabla A^*(\mu_{\mathrm{MAP}}^{(n)})$, the MAP writes:

$$\nabla A(\theta^{(n+1)}) = \nabla A(\theta^{(n)}) - \eta \nabla f_{X_n}(\theta^{(n)}), \tag{12}$$

where $f_{X_n}(\theta) = A(\theta) - \langle \theta, T(X_n)\rangle$, so that $f(\theta) = A(\theta) - \langle \theta, \mu_\star\rangle$. We recognize stochastic mirror descent iterations, with mirror $A$ and stochastic gradients $\nabla f_X$. Similar results on the MLE can be obtained by taking $n_0 = 0$. This key observation implies that **convergence guarantees on the MAP and the MLE can be deduced from stochastic mirror descent convergence guarantees**.

While this appears as an appealing way to obtain convergence guarantees for the MAP, Le Priol et al. [17] observe that none of the existing SMD results obtain meaningful rates for the convergence of the MAP for general exponential families. In particular, none of them recover the $O(1/n)$ asymptotic convergence rate for estimating a Gaussian with unknown mean and covariance.

This is due to the variance definitions used in the existing analyses, that all have issues (not uniformly bounded over the domain, not decreasing with the step-size...) as discussed in Section 2. Our analysis fixes this problem, and thus yields finite-time guarantees for the MAP estimator for the estimation of a Gaussian with unknown mean and covariance. This shows the relevance of Assumption 2.

### 4.2  Full Gaussian (unknown mean and covariance)

The main problem studied in Le Priol et al. [17] is that of the one-dimensional full-Gaussian case, where the goal is to estimate the mean and covariance of a Gaussian from i.i.d. samples $X_1, \ldots, X_n \sim \mathcal{N}(m_\star, \Sigma_\star)$, with $\Sigma_\star > 0$. Note that although notation $\Sigma$ is usually reserved for the covariance matrix of a multivariate Gaussian, we use it for a scalar value here to highlight the distinction with $\sigma_{\star,\eta}^2$, the variance from stochastic mirror descent. In this case, the sufficient statistics write $T(X) = (X, X^2)$, and the log-partition and entropy functions are, up to constants, $A(\theta) = \frac{\theta_1^2}{-4\theta_2} - \frac{1}{2}\log(-\theta_2), A^*(\mu) = -\frac{1}{2}\log(\mu_2 - \mu_1^2)$, for $\theta \in \Theta = \mathbb{R} \times \mathbb{R}_-^*$ and $\mu \in \{(u,v), u^2 < v\}$. The goal is to estimate $D_A(\theta, \theta_\star)$, for which Le Priol et al. [17] show that only partial solutions exist: results are either asymptotic, or rely on the objective being (approximately) quadratic. Note that there is a relationship between natural parameters, mean parameters, and $(m, \Sigma^2)$, the mean and covariance of the Gaussian we would like to estimate. In the following, we will often abuse notations, and write for instance $D_A(\tilde{\theta}, \theta)$ in terms of $(m, \Sigma^2)$ and $(\tilde{m}, \tilde{\Sigma}^2)$ rather than $\theta$ and $\tilde{\theta}$. We now state a few results, for which detailed derivations can be found in Appendix F. More specifically:

$$D_A(\tilde{\theta}, \theta) = -\frac{1}{2}\log\left(\frac{\Sigma^2}{\tilde{\Sigma}^2}\right) - \frac{\tilde{\Sigma}^2 - \Sigma^2}{2\tilde{\Sigma}^2} + \frac{(\tilde{m} - m)^2}{2\tilde{\Sigma}^2}.$$

The update formulas for the parameters are given by:
$$m^+ = (1 - \eta)m + \eta X, \qquad (\Sigma^2)^+ = (1 - \eta)\left[\Sigma^2 + \eta(m - X)^2\right]. \tag{13}$$
Therefore, MAP iterations are well-defined although $A$ does not verify Assumption 1.

**Proposition 4.1.** *The iterations* (12) *are well-defined for $\eta < 1$ in the sense that if $\theta^{(n)} \in \Theta = \mathbb{R} \times \mathbb{R}_-^*$,*
*then $\nabla A(\theta^{(n)}) - \eta \nabla f_{X_n}(\theta^{(n)}) \in \mathrm{Range}(\nabla A)$ almost surely, so that $\theta^{(n+1)} \in \Theta$ is well-defined*
*almost surely. Besides, $f_\xi$ is 1-relatively-smooth and 1-relatively-strongly-convex with respect to $A$.*

This result is a direct consequence of the fact that $D_{f_\xi} = D_f = D_A$ for all $\xi$, and the fact that
$\nabla A(\theta) - \eta \nabla f_{X_n}(\theta) = (1 - \eta)\nabla A(\theta) + \eta T(X_n) \in \{(u, v), u^2 < v\}$ if $\nabla A(\theta) \in \{(u, v), u^2 < v\}$.
Proposition 4.1 means that we can apply Theorem 3.1, so the next step is to bound the variance $\sigma_{\star,\eta}^2$.

$$f_\eta(\theta) - f(\theta_\star) = \frac{1}{2\eta}\mathbb{E}\left[\log\left((1 - \eta)\left(1 + \eta\frac{(m - X)^2}{\Sigma^2}\right)\right)\right] - \frac{1}{2}\log\left(\frac{\Sigma_\star^2}{\Sigma^2}\right). \tag{14}$$

We now use this expression to to lower bound $f_\eta^\star$ and so upper bound $\sigma_{\star,\eta}^2$.

**Lemma 4.2.** *Let $(m_\eta, \Sigma_\eta^2)$ be the minimizer of $f_\eta$. Then, for $\eta < 1/3$, $m_\eta = m_\star$, $\Sigma_\star^2 \geq \Sigma_\eta^2 \geq$*
*$(1 - 3\eta)\Sigma_\star^2$. In particular, the variance $\sigma_{\star,\eta}^2$ verifies $\sigma_{\star,\eta}^2 \leq -\frac{1}{2\eta}\log(1 - 3\eta)$. For $1/3 < \eta \leq 1 - \varepsilon$,*
*$\sigma_{\star,\eta}^2 \leq c_\varepsilon$, where $c_\varepsilon$ is a numerical constant that only depends on $\varepsilon$.*

Note that we show in this example that $\Sigma_\eta^2$ is arbitrarily close to $\Sigma_\star^2$ as $\eta \to 0$, which is expected.

**Theorem 4.3.** *Let $\Gamma \geq 0$ be a numerical constant and $\Gamma = 0$ if $n_0 > 3$. The MAP estimator satisfies:*

$$\mathbb{E}\left[D_A(\theta_\star, \theta^{(n)})\right] \leq \frac{n_0 D_A(\theta_\star, \theta^{(0)}) + \frac{3}{2}\log(1 + \frac{n+1}{n_0}) + \Gamma}{n + n_0}.$$

Numerical constants are not optimized. Theorem 4.3 gives an anytime result on the convergence of
the MAP estimator for all $n \geq 0, n_0 \geq 1$ directly from the general SMD convergence theorem. Yet,
the open problem from Le Priol et al. [17] is not completely solved still, as discussed below.

**Reverse KL bound.** We obtain a bound on $D_A(\theta_\star, \theta^{(n)})$, instead of $D_A(\theta^{(n)}, \theta_\star) = f(\theta) - f(\theta_\star)$.
$D_A(\theta^{(n)}, \theta_\star)$ can be controlled asymptotically thanks to the bound on $f_\eta(\theta^{(n)}) - f_\eta(\theta_\eta)$, and $f_\eta \to f$
when $\eta = 1/n \to 0$, but we might also be able to exploit this control over the course of the iterations.

**Asymptotic convergence.** Theorem 4.3 leads to a $O(\log n/n)$ asymptotic convergence rate instead
of the expected $O(1/n)$ [17]. This indicates that the $f_{\eta_n}(\theta^{(n)}) - f_{\eta_n}^\star$ terms should not be neglected.
Indeed, $\theta^{(n)}$ actually has a lot of structure in this example, since $\nabla A(\theta^{(n)}) = \frac{1}{n}\sum_{k=1}^n T(X_k)$. The
SMD analysis is oblivious to this structure, hence the gap. Note that we can get rid of the $\log n$ factor
and recover the right $O(1/n)$ rate from the same analysis by using a slightly different estimator than
the MAP (or MLE). This is done by setting the step-size as $\eta_n = \frac{2}{n+1}$ for $n > 1$, and the analysis of
this variant follows Lacoste-Julien et al. [16], as detailed in Appendix F.3.

**The special case of the MLE.** The MLE corresponds to $n_0 = 0$, which is not handled in our analysis
since the first step corresponds to $\eta = 1$, which necessarily results in $\theta_2^{(1)} = -\infty$ (which corresponds
to $\Sigma^2 = 0$, as can be seen from (13)). If we consider that mirror descent is run from $\theta^{(1)}$, then we
obtain $\mathbb{E}\left[D_A(\theta_\star, \theta^{(1)})\right] = \infty$ in general, where the expectation is over the value of the first sample
drawn. Therefore, we need to start the SMD analysis at $\theta^{(2)}$ to fit the MLE into this framework, and in
particular we need to be able to evaluate $\mathbb{E}\left[D_A(\theta_\star, \theta^{(2)})\right]$. This is further discussed in Appendix F.4.

## 5    Conclusion

This paper introduces a new notion of variance for the analysis of stochastic mirror descent. This
notion, based on the fact that a certain function $f_\eta$ admits a minimum, is less restrictive than existing
ones, has the right asymptotic scaling with the step-size and is bounded regardless of the trajectory of
the iterates without further assumptions.

We strongly believe that our analysis of SMD opens up new perspectives. As an example, we use our
SMD results to show convergence of the MAP for estimating a Gaussian with unknown mean and
covariance. As evidenced in Le Priol et al. [17], all existing generic analyses of stochastic mirror
descent failed to obtain such results.

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

# A Technical results on Bregman divergences

As for the rest of this paper, Assumption 1 is assumed throughout this section. However, some of these results hold even with less regularity, and in particular do not require second order continuous differentiability.

**Lemma A.1** (Duality). *For all $x, y \in C$, it holds that:*

$$D_h(x, y) = D_{h^*}(\nabla h(y), \nabla h(x)) \tag{15}$$

See, *e.g.* Bauschke et al. [1, Theorem 3.7] for the proof.

**Lemma A.2** (Symmetrized Bregman). *For all $x, y \in C$, it holds that:*

$$D_h(x, y) + D_h(y, x) = \langle \nabla h(x) - \nabla h(y), x - y \rangle \tag{16}$$

The proof immediately follows from the definition of the Bregman divergence. The following result corresponds to Dragomir et al. [7, Lemma 3].

**Lemma A.3** (Bregman cocoercivity). *If a convex function $f$ is $L$-relatively-smooth with respect to $h$, then for all $\eta \leq 1/L$,*

$$D_{h^*}(\nabla h(x) - \eta [\nabla f(x) - \nabla f(y)], \nabla h(x)) \leq \eta D_f(x, y). \tag{17}$$

*Denoting $x^{+y} = \nabla h^*(\nabla h(x) - \eta [\nabla f(x) - \nabla f(y)])$, a tighter result actually writes:*

$$D_h(x, x^{+y}) + \eta D_f(x^{+y}, y) \leq \eta D_f(x, y). \tag{18}$$

The proof of the tighter version is simply obtained by not using that $D_f(x^{+y}, y) \geq 0$ in the original proof. While we don't directly use it in this paper, it is sometimes useful. We now introduce the generalized bias-variance decomposition Lemma [23, Theorem 0.1].

**Lemma A.4.** *If $X$ is a random variable, then for all $u \in C$,*

$$\mathbb{E}[D_{h^*}(X, u)] = D_{h^*}(\mathbb{E}[X], u) + D_{h^*}(X, \mathbb{E}[X]). \tag{19}$$

# B Missing results on the variances

We start this section by proving the following lemma, which in particular ensures that $D_h(x, x^+)/\eta$ increases with $\eta$ (and so decreases as $\eta \to 0$).

**Lemma B.1.** *Let $\phi_\xi : \eta \mapsto \frac{1}{\eta} D_h(x, x^+(\eta, \xi))$. Then, $\nabla \phi_\xi(\eta) = \frac{1}{\eta^2} D_h(x^+(\eta, \xi), x) \geq 0$.*

*Proof.* First remark that since $\nabla h(x^+) = \nabla h(x) - \eta \nabla f_\xi(x)$, we can write

$$
\begin{aligned}
\nabla_\eta [D_h(x, x^+)] &= \nabla_\eta [h(x) - h(x^+) - \nabla h(x^+)^\top (x - x^+)] \\
&= -\nabla h(x^+)^\top \nabla_\eta x^+ + \nabla f_\xi(x)^\top (x - x^+) + \nabla h(x^+)^\top \nabla_\eta x^+ \\
&= \nabla f_\xi(x)^\top (x - x^+) \\
&= \frac{1}{\eta} (\nabla h(x) - \nabla h(x^+))^\top (x - x^+) = \frac{D_h(x, x^+) + D_h(x^+, x)}{\eta}.
\end{aligned}
$$

Then, the expression follows from

$$\nabla \phi_\xi(\eta) = \nabla_\eta \left[ \frac{1}{\eta} D_h(x, x^+) \right] = \frac{1}{\eta} \nabla_\eta [D_h(x, x^+)] - \frac{1}{\eta^2} D_h(x, x^+) = \frac{1}{\eta^2} D_h(x^+, x). \tag{20}$$

$\square$

*Proof of Proposition 2.5.* We now prove that $f_\eta \to f$ when $\eta \to 0$. To show this, we note that for any fixed $x \in \text{int } C$:

- For any fixed $\xi$, $\frac{1}{\eta} D_h(x, x^+) = \frac{\eta}{2} \|\nabla f_\xi(x)\|^2_{\nabla^2 h^*(z)}$ for $z \in [\nabla h(x), \nabla h(x) - \eta \nabla f_\xi(x)]$. Therefore, $\frac{1}{\eta} D_h(x, x^+) \to 0$ for $\eta \to 0$ since $\nabla^2 h^*(\nabla h(x)) = (\nabla^2 h(x))^{-1} < \infty$ by strict convexity of $h$.

- Let $\eta \leq \eta_0$. Then, for all $\xi$, $\frac{1}{\eta} D_h(x, x^+(\eta, \xi)) \leq \frac{1}{\eta_0} D_h(x, x^+(\eta_0, \xi))$ since the function $\eta \mapsto \frac{1}{\eta} D_h(x, x^+(\eta, \xi))$ is an increasing function (positive gradient using Lemma B.1).

- $\frac{1}{\eta_0} \mathbb{E}\left[D_h(x, x^+(\eta_0, \xi))\right]$ is finite.

Then, using the dominated convergence theorem, we obtain that we can invert the integral (expectation) and the limit, so that $\lim_{\eta \to 0} \mathbb{E}\frac{1}{\eta} D_h(x, x^+) = \mathbb{E} \lim_{\eta \to 0} \frac{1}{\eta} D_h(x, x^+) = 0$. $\qquad \square$

*Proof of Proposition 2.6.* We prove this result by successively upper bounding and lower bounding $\sigma^2_{\star,\eta}$, and making $\eta \to 0$.

*1 - Upper bound on $\sigma^2_{\star,\eta}$.* One side is direct, by writing that $f(x_\eta) \geq f(x_\star)$:

$$\sigma^2_{\star,\eta} = \frac{1}{\eta}\left(f(x_\star) - f(x_\eta) + \frac{1}{\eta}\mathbb{E}\left[D_h(x_\eta, x_\eta^+)\right]\right) \leq \frac{1}{\eta^2}\mathbb{E}\left[D_h(x_\eta, x_\eta^+)\right]. \tag{21}$$

From the proof of Proposition 2.5 we have pointwise convergence of $f_\eta$ to $f$. Since $f$ is convex and has a unique minimizer $x_\star$, then $x_\eta \to x_\star$ for $\eta \to 0$, which leads to the result.

*2 - Lower bound on $\sigma^2_{\star,\eta}$.* By definition of $x_\eta$ as the minimizer of $f_\eta$, we have $f_\eta(x_\eta) \leq f_\eta(x_\star)$, and so:

$$\sigma^2_{\star,\eta} = \frac{f(x_\star) - f_\eta(x_\eta)}{\eta} \geq \frac{f(x_\star) - f_\eta(x_\star)}{\eta} = \frac{1}{\eta^2}\mathbb{E}\left[D_h(x_\star, x_\star^+)\right]. \tag{22}$$

$\qquad \square$

Let us now prove the following proposition, which follows the proof from Dragomir et al. [7].

*Proof of Proposition 2.7.* Let us prove that $\sigma^2_{\star,\eta} \leq \mathbb{E}\left[\|\nabla f_\xi(x_\star)\|^2_{\nabla^2 h^*(z_\eta)}\right]$. We start by

$$D_h(x, x^+) = D_{h^*}(\nabla h(x) - \eta\nabla f_\xi(x), \nabla h(x)) \tag{23}$$
$$= D_{h^*}(\nabla h(x) - \eta\left[\nabla f_\xi(x) - \nabla f_\xi(x_\star)\right] - \eta\nabla f_\xi(x_\star), \nabla h(x)) \tag{24}$$
$$= D_{h^*}(\frac{(\nabla h(x) - 2\eta\left[\nabla f_\xi(x) - \nabla f_\xi(x_\star)\right]) + (\nabla h(x) - 2\eta\nabla f_\xi(x_\star))}{2}, \nabla h(x)). \tag{25}$$

Using the convexity of $D_{h^*}$ in its first argument and then the Bregman cocoercivity lemma, we obtain for $\eta \leq 1/2L$:

$$D_h(x, x^+) \leq \frac{1}{2}D_{h^*}(\nabla h(x) - 2\eta\left[\nabla f_\xi(x) - \nabla f_\xi(x_\star)\right], \nabla h(x)) \tag{26}$$
$$+ \frac{1}{2}D_{h^*}(\nabla h(x) - 2\eta\nabla f_\xi(x_\star), \nabla h(x)) \tag{27}$$
$$\leq \eta D_{f_\xi}(x, x_\star) + \frac{1}{2}D_{h^*}(\nabla h(x) - 2\eta\nabla f_\xi(x_\star), \nabla h(x)). \tag{28}$$

Using that $\mathbb{E}\left[D_{f_\xi}(x, x_\star)\right] = D_f(x, x_\star)$ and applying this to $x = x_\eta$, we obtain

$$\sigma^2_{\star,\eta} = \frac{f(x_\star) - f_\eta(x_\eta)}{\eta}$$
$$= \frac{\mathbb{E}\left[D_h(x_\eta, x_\eta^+)\right] + \eta f(x_\star) - \eta f(x_\eta)}{\eta^2}$$
$$\leq \frac{1}{2\eta^2}\mathbb{E}\left[D_{h^*}(\nabla h(x_\eta) - 2\eta\nabla f_\xi(x_\star), \nabla h(x_\eta))\right] + \frac{D_f(x_\eta, x_\star) + f(x_\star) - f(x_\eta)}{\eta}$$
$$= \frac{1}{2\eta^2}\mathbb{E}\left[D_{h^*}(\nabla h(x_\eta) - 2\eta\nabla f_\xi(x_\star), \nabla h(x_\eta))\right] = \frac{1}{2\eta^2} \times \mathbb{E}\left[\frac{1}{2}\|2\eta\nabla f_\xi(x_\star)\|^2_{\nabla^2 h^*(z_\eta)}\right],$$

and the result follows. The last inequality comes from the fact that if $x_\star = \arg\min_{x \in C} f(x)$, then $-\nabla f(x_\star)$ is normal to $C$ so $-\nabla f(x_\star)^\top(x_\eta - x_\star) \leq 0$. . $\qquad \square$

## C  Convergence results.

In this section, we detail the proofs of the various convergence theorems that were only sketched in the main text. We start by proving the first identity, which is a variation of *e.g.*, Dragomir et al. [7, Lemma 4], which we detail here for the sake of completeness.

**Lemma C.1.** *Let $x^+ \in C$ be such that $\nabla h(x^+) = \nabla h(x) - \eta \nabla f_\xi(x)$, with $f_\xi$ a random differentiable function such that $\mathbb{E}[f_\xi] = f$. Then, for all points $y \in C$,*

$$\mathbb{E}\left[D_h(y, x^+)\right] - D_h(y, x) + \eta D_f(y, x) = -\eta[f(x) - f(y)] + \mathbb{E}\left[D_h(x, x^+)\right] \tag{29}$$

*In particular, we can apply the result to $y = x_\star$.*

*Proof.* We give a slightly different proof than Dragomir et al. [7], and in particular this version of the identity is slightly more direct (though maybe less insightful) and does not require $\nabla f(y) = 0$. We write:

$$\begin{aligned}
\mathbb{E}\left[D_h(y, x^+)\right] &= \mathbb{E}\left[h(y) - h(x^+) - \nabla h(x^+)^\top (y - x^+)\right] \\
&= \mathbb{E}\left[h(y) - h(x^+) - \nabla h(x^+)^\top (y - x) - \nabla h(x^+)^\top (x - x^+)\right] \\
&= \mathbb{E}\big[h(y) - h(x) - \nabla h(x)^\top (y - x) + \eta \nabla f_\xi(x)^\top (y - x) \\
&\qquad \nabla h(x^+)^\top (x - x^+) + h(x) - h(x^+)\big] \\
&= D_h(y, x) + \eta \nabla f(x)^\top (y - x) + \mathbb{E}\left[D_h(x, x^+)\right] \\
&= D_h(y, x) - \eta D_f(y, x) + \eta \left[f(y) - f(x)\right] + \mathbb{E}\left[D_h(x, x^+)\right].
\end{aligned}$$

$\square$

*Proof of Corollary 3.2.* We start back from Equation (8), and write, using that $f_\eta(x) \geq \mathbb{E}[f_\xi(x^+)]$ (proof of Proposition 2.2):

$$\mathbb{E}\left[D_h(x_\star, x^+)\right] - D_h(x_\star, x) + \eta D_f(x_\star, x) = \eta\left[f(x_\star) - f_\eta(x)\right] \tag{30}$$

$$\leq \eta\left[f(x_\star) - \mathbb{E}\left[f_\xi(x^+)\right]\right] \tag{31}$$

$$\leq -\eta\left[\mathbb{E}\left[f_\xi(x^+)\right] - f_\star^+\right] + \eta^2\left(\frac{f(x_\star) - f_\star^+}{\eta}\right). \tag{32}$$

The result follows naturally from using the relative strong convexity of $f$, leading to:

$$\eta[\mathbb{E}\left[f_\xi(x^+)\right] - f_+^\star] + \mathbb{E}\left[D_h(x_\star, x^+)\right] \leq (1 - \eta\mu)D_h(x_\star, x) + \eta^2\left[\frac{f(x_\star) - f_+^\star}{\eta}\right]. \tag{33}$$

Then, we chain iterations as done for Theorem 3.1

$\square$

*Proof of Theorem 3.3.* We also start from the same result as above, and write it for $x = x^{(k)}$, so that $x^+ = x^{(k+1)}$:

$$\mathbb{E}\left[D_h(x_\star, x^{(k+1)})\right] - D_h(x_\star, x^{(k)}) + \eta D_f(x_\star, x^{(k)}) = \eta\left[f(x_\star) - f_\eta(x^{(k)})\right] \tag{34}$$

$$\leq -\eta\left[f_\eta(x^{(k)}) - f_\eta(x_\eta)\right] + \eta^2 \sigma_{\star,\eta}^2. \tag{35}$$

Moving the $f_\eta$ terms to the left, and summing this for $k = 0$ to $T$ leads to:

$$\eta \sum_{k=0}^{T}\left[f_\eta(x^{(k)}) - f_\eta(x_\eta) + D_f(x_\star, x^{(k)})\right] \leq D_h(x_\star, x^{(0)}) - \mathbb{E}\left[D_h(x_\star, x^{(k+1)})\right] + T\eta^2 \sigma_{\star,\eta}^2. \tag{36}$$

The final result is obtained by dividing by $\eta T$, and the fact that $\mathbb{E}\left[D_h(x_\star, x^{(k+1)})\right] \geq 0$.

$\square$

*Proof of Proposition 3.4.* We use Bregman cocoercivity (Lemma A.3) with $\eta = \frac{1}{L}$ between $x_\star$ and $x$ (instead of $x$ and $x_\star$ as it had been done previously), which directly leads to:

$$D_{h^*}\left(\nabla h^*(x_\star) - \frac{1}{L}\left[\nabla f(x_\star) - \nabla f(x)\right]\right) \leq \frac{1}{L}D_f(x_\star, x). \tag{37}$$

The first part of the proposition follows from the fact that $\nabla f(x_\star) = 0$. For the rest proof, we start with Inequality (32), which gives:

$$0 = D_{h^*}(\nabla h(x_\star) - \frac{1}{L}\nabla f(x), \nabla h(x_\star))$$

$$= D_h\left(x_\star, \nabla h^*\left(\nabla h(x_\star) - \frac{1}{L}\nabla f(x)\right)\right).$$

At this point, strict convexity of $h$ leads to $\nabla h^*\left(\nabla h(x_\star) - \frac{1}{L}\nabla f(x)\right) = x_\star$, so that $\nabla f(x) = 0$ by applying $\nabla h$ on both sides. $\qquad\square$

## D   Variation on the convex case

In this section, we quickly illustrate that the result we obtain is tightly linked to the notion of variance that we define. As an example, a variation of Theorem 3.3 can be obtained with a control on $f(x) - f(x_\star)$, but this requires a different notion of variance:

**Theorem D.1.** *If $f$ is convex, the iterates obtained by SMD using a constant step-sizes $\eta > 0$ verify*

$$\mathbb{E}\left[f\left(\frac{1}{T}\sum_{k=0}^{T} x^{(k)}\right)\right] - f(x_\star) \leq \frac{D_h(x_\star, x^{(0)})}{\eta T} + \eta\tilde{\sigma}_{\star,\eta}^2, \tag{38}$$

*where*

$$\tilde{\sigma}_{\star,\eta}^2 = \frac{1}{\eta}\max_{x\in C}\left\{\frac{1}{\eta}\mathbb{E}\left[D_h(x, x^+)\right] - D_f(x_\star, x)\right\}. \tag{39}$$

Note that this alternative variance definition can be unbounded even when $\sigma_{\star,\eta}^2$ is bounded, as is the case for instance in the Gaussian MAP example. Besides, it does not inherit from most of the good properties of $\sigma_{\star,\eta}^2$ presented in Section 2, and cannot be compared to the other standard variance notions. The main case in which this alternative definition makes sense is the Euclidean case, in which $\tilde{\sigma}_{\star,\eta}^2$ can be bounded using cocoercivity.

*Proof of Theorem D.1.* This proof directly starts from Lemma C.1:

$$\mathbb{E}\left[D_h(x_\star, x^{(k+1)})\right] \tag{40}$$

$$= D_h(x_\star, x^{(k)}) - \eta D_f(x_\star, x^{(k)}) - \eta[f(x^{(k)}) - f(x_\star)] + \mathbb{E}\left[D_h(x^{(k)}, (x^{(k)})^+)\right] \tag{41}$$

$$= D_h(x_\star, x^{(k)}) - \eta[f(x^{(k)}) - f(x_\star)] + \eta\left[\frac{1}{\eta}\mathbb{E}\left[D_h(x^{(k)}, (x^{(k)})^+)\right] - D_f(x_\star, x^{(k)})\right] \tag{42}$$

$$\leq D_h(x_\star, x^{(k)}) - \eta[f(x^{(k)}) - f(x_\star)] + \eta^2\tilde{\sigma}_{\star,\eta}^2. \tag{43}$$

Summing this for $k = 0$ to $T$, and dividing by $\eta T$ we obtain:

$$\frac{1}{T}\sum_{k=0}^{T} f(x^{(k)}) - f(x_\star) \leq \frac{D_h(x_\star, x^{(0)})}{\eta T} + \eta\tilde{\sigma}_{\star,\eta}^2 \tag{44}$$

The result on the average iterate then follows from convexity of $f$ and taking expectation on $x^{(k)}$. $\quad\square$

## E   Stochastic Mirror Descent with a Proximal term

We are interested in this section in a variation of the original problem, where we would like to solve the following problem:

$$\min_{x\in C} f(x) + g(x), \tag{45}$$

where $g$ is a convex proper lower semi-continuous function (but not necessarily differentiable). This problem can be solved using the following stochastic proximal mirror descent algorithm:

$$x^+ = \arg\min_{u\in C} g(u) + \nabla f_\xi(x)^\top u + \frac{1}{\eta}D_h(u, x). \tag{46}$$

This is a "proximal" version, which for instance corresponds to projected stochastic mirror descent if $g$ is the indicator of a convex set. Under Assumption 1, the iterations write:

$$\nabla h(x^+) = \nabla h(x) - \eta \left[ \nabla f_\xi(x) + \omega \right] \tag{47}$$

where $\omega \in \partial g(x^+)$, the subgradient of $g$ at point $x^+$. Equation (47) can be rewritten as

$$\nabla h(x^+) + \eta \omega = \nabla h(x) + \eta \omega_x - \eta \left[ \nabla f_\xi(x) + \omega_x \right] \tag{48}$$

for any $\omega_x \in \partial g(x)$. In particular, (47) can be interpreted as a Stochastic mirror descent step with objective $f_\xi + g$ and mirror $h + \eta g$. While the mirror does not satisfy Assumption 1 (and in particular twice differentiability in case $g$ is the indicator of a set), the iterations can still be written in the form of Equation (3). In particular, the theorems from Section 3 still apply, with the adapted variance definition involving function $f + g$ and mirror $h + \eta g$. Similarly, $f + g$ is $1/\eta$ relatively-smooth with respect to $h + \eta g$ as long as $f$ is $L$-relatively-smooth with respect to $h$ and $\eta \leq 1/L$.

We now prove an equivalent for Lemma C.1.

**Lemma E.1.** *Let $x^+ \in C$ be such that $\nabla h(x^+) = \nabla h(x) - \eta \left[ \nabla f_\xi(x) + \omega \right]$, with $f_\xi$ a random differentiable function such that $\mathbb{E}\left[ f_\xi \right] = f$ and $\omega \in \partial g(x^+)$ where $g$ is a convex proper lower semi-continuous function. Then, for all $y \in C \cap \mathrm{dom} g$,*

$$\mathbb{E}\left[ D_h(y, x^+) \right] = D_h(y, x) - \eta D_f(y, x) - \eta[f(x) - f(y)]$$
$$+ \mathbb{E}\left[ D_h(x, x^{+f}) - D_h(x^+, x^{+f}) \right] + \eta \omega^\top (y - x^+),$$

*where $x^{+f}$ is the point such that $\nabla h(x^{+f}) = \nabla h(x) - \eta \nabla f_\xi(x)$.*

*Proof.* We write:

$$D_h(y, x^+) = h(y) - h(x^+) - \nabla h(x^+)^\top (y - x^+)$$
$$= h(y) - h(x^{+f}) - \nabla h(x^{+f})^\top (y - x^+) + \eta \omega^\top (y - x^+) - h(x^+) + h(x^{+f})$$
$$= D_h(y, x^{+f}) - \nabla h(x^{+f})^\top (x^{+f} - x^+) + \eta \omega^\top (y - x^+) - h(x^+) + h(x^{+f})$$
$$= D_h(y, x^{+f}) - D_h(x^+, x^{+f}) + \eta \omega^\top (y - x^+)$$

The result follows from applying Lemma C.1 to $D_h(y, x^{+f})$. $\qquad\square$

Note that by abuse of notation, if we denote $D_g(y, x^+) = g(y) - g(x^+) - \omega^\top (y - x^+)$, and $D_g(y, x) = g(y) - g(x) - \omega_x^\top (y - x)$ for any $\omega_x \in \partial g(x)$, then with a few lines of computations, and noting in particular that $D_h(x, x^{+f}) - D_h(x^+, x^{+f}) = D_h(x, x^+) - \left[ \nabla h(x^{+f}) - \nabla h(x^+) \right]^\top (x - x^+)$ we obtain:

$$\mathbb{E}\left[ D_{h+\eta g}(y, x^+) \right] = D_h(y, x) - \eta D_f(y, x) - \eta[f(x) - f(y)]$$
$$+ \mathbb{E}\left[ D_h(x, x^+) \right] - \eta \mathbb{E}\left[ \omega^\top (x - x^+) \right] + \eta \mathbb{E}\left[ g(y) - g(x^+) \right]$$
$$= D_{h+\eta g}(y, x) - \eta D_g(y, x) - \eta D_f(y, x) - \eta[f(x) - f(y)]$$
$$+ \mathbb{E}\left[ D_{h+\eta g}(x, x^+) \right] + \eta \left[ g(y) - g(x) \right]$$

In particular, we exactly recover the result of Lemma C.1 applied to the iterations in which we take (sub)-gradients of $f + g$ with mirror $h + \eta g$, *i.e.*,

$$\mathbb{E}\left[ D_{h+\eta g}(y, x^+) \right] = D_{h+\eta g}(y, x) - \eta D_{f+g}(y, x) - \eta[(f + g)(x) - (f + g)(y)] + \mathbb{E} D_{h+g}(x, x^+).$$

Therefore, using the same sequence of derivations, Theorems 3.1 and 3.3 can be transposed directly to the composite $(f + g)$ setting by simply defining generalized Bregman divergences where the gradient parts are replaced by the subgradients picked in the actual SMD steps.

While $h + \eta g$ does not necessarily satisfy Assumption 1, the key point is that iterations can be written in the form of Equation (47), which is the case for instance if $g$ is the indicator of a convex set.

Note that Corollary 3.2 also holds in the same way, since relative smoothness is only needed to obtain that $\eta D_{f_\xi + g}(x, x^+) \leq D_{h+\eta g}(x, x^+)$, which is equivalent to $\eta D_{f_\xi}(x, x^+) \leq D_h(x, x^+)$, which holds by $L$-relative smoothness of $f$ with respect to $h$ for $\eta \leq 1/L$.

 # F   Gaussian case with unknown covariance.

 In this section, we prove the various results for Gaussian estimation with unknown mean and
 covariance. For the sake of brevity, we only prove the propositions, and refer the interested reader to,
 *e.g.*, Le Priol et al. [17] for standard results about the setting.

 ## F.1   Instanciation in the Stochastic mirror descent setting

 We first write what the various divergences are in our setting, together with the mirror updates and
 finally the form of $f_\eta$. Following Le Priol et al. [17, Section 4.2], we write that:

$$\theta_1 = \frac{m}{\Sigma^2}, \qquad \theta_2 = -\frac{1}{2\Sigma^2}. \tag{49}$$

 This allows us to express $A(\theta)$ in terms of $(m, \Sigma^2)$:

$$A(\theta) = -\frac{1}{2}\log(-\theta_2) - \frac{\theta_1^2}{4\theta_2} = \frac{1}{2}\log(2\Sigma^2) + \frac{1}{2}\frac{m^2}{\Sigma^2} \tag{50}$$

 **Proposition F.1.** *The Bregman divergence with respect to $\tilde{\theta}, \theta$ writes:*

$$D_A(\tilde{\theta}, \theta) = -\frac{1}{2}\log\left(\frac{\Sigma^2}{\tilde{\Sigma}^2}\right) - \frac{\tilde{\Sigma}^2 - \Sigma^2}{2\tilde{\Sigma}^2} + \frac{(\tilde{m} - m)^2}{2\tilde{\Sigma}^2}. \tag{51}$$

 *Proof.* We know that $\nabla A(\theta) = \mu = (m, m^2 + \Sigma^2)$. Therefore,

$$\nabla A(\theta)^\top(\tilde{\theta} - \theta) = m\left(\frac{\tilde{m}}{\tilde{\Sigma}^2} - \frac{m}{\Sigma^2}\right) - \frac{1}{2}(m^2 + \Sigma^2)\left(\frac{1}{\tilde{\Sigma}^2} - \frac{1}{\Sigma^2}\right) \tag{52}$$

$$= \frac{m\tilde{m}}{\tilde{\Sigma}^2} - \frac{m^2}{2\Sigma^2} - \frac{m^2}{2\tilde{\Sigma}^2} - \frac{1}{2}\left(\frac{\Sigma^2}{\tilde{\Sigma}^2} - 1\right) \tag{53}$$

$$= -\frac{(m - \tilde{m})^2}{2\tilde{\Sigma}^2} + \frac{\tilde{m}^2}{2\tilde{\Sigma}^2} - \frac{m^2}{2\Sigma^2} - \frac{\Sigma^2 - \tilde{\Sigma}^2}{2\tilde{\Sigma}^2}. \tag{54}$$

 Using Equation (50), we obtain:

$$D_A(\tilde{\theta}, \theta) = A(\tilde{\theta}) - A(\theta) - \nabla A(\theta)^\top(\tilde{\theta} - \theta)$$
$$= \frac{1}{2}\log(2\tilde{\Sigma}^2) - \frac{1}{2}\log(2\Sigma^2) + \frac{\Sigma^2 - \tilde{\Sigma}^2}{2\tilde{\Sigma}^2} + \frac{(m - \tilde{m})^2}{2\tilde{\Sigma}^2},$$

 which finishes the proof. $\qquad\square$

 In the Gaussian with unknown covariance, the sufficient statistics are:

$$T(X) = (X, X^2), \tag{55}$$

 where $x \in \mathbb{R}$ is an observation drawn from $\mathcal{N}(m_\star, \Sigma_\star)$.

 Let us now prove the form on the updates, which corresponds to (13):

 **Proposition F.2.** *In $(m, \Sigma^2)$ parameters, the updates write:*

$$m^+ = (1 - \eta)m + \eta X, \tag{56}$$
$$(\Sigma^2)^+ = (1 - \eta)\left[\Sigma^2 + \eta(m - X)^2\right]. \tag{57}$$

 *Proof.* Since the (stochastic) gradients write $g(\mu) = \mu - T(X)$, the iterations are defined by:

$$\mu_1^+ = (1 - \eta)\mu_1 + \eta X \tag{58}$$
$$\mu_2^+ = (1 - \eta)\mu_2 + \eta X^2. \tag{59}$$

579  Since $(\mu_1, \mu_2) = (m, m^2 + \Sigma^2)$, the update on $m$ is immediate. For the update on $\Sigma^2$, we write:

$$\begin{aligned}
(\Sigma^2)^+ &= \mu_2^+ - (m^+)^2 \\
&= (1-\eta)\mu_2 + \eta X^2 - ((1-\eta)m + \eta X)^2 \\
&= (1-\eta)\Sigma^2 + (1-\eta)m^2 + \eta X^2 - (1-\eta)^2 m^2 - 2\eta(1-\eta)Xm - \eta^2 X^2 \\
&= (1-\eta)\Sigma^2 + \eta(1-\eta)(m-X)^2.
\end{aligned}$$

580  $\qquad\square$

581  We now use this to show that updates are well-defined.

582  *Proof of Proposition 4.1.* If $\theta_2 < 0$ then $\Sigma^2 > 0$ so for $\eta < 1$, $(\Sigma^2)^+ > 0$ almost surely so that
583  $\theta_2^+ < 0$ and $|\theta_1^+| < \infty$. In particular, $\theta^+ \in \mathbb{R} \times \mathbb{R}_*^*$ so the update is well-defined. The rest of the
584  proposition comes from the fact that $\nabla^2 f_\xi = \nabla^2 f = \nabla^2 A$. $\qquad\square$

585  We can now proceed to proving the form of $f_\eta$. We first start by writing that:

$$f(\theta) = A(\theta) - \theta^\top(m_\star, m_\star^2 + \Sigma_\star^2) \tag{60}$$

$$= \frac{1}{2}\log(2\Sigma^2) + \frac{1}{2}\frac{m^2}{\Sigma^2} - \frac{mm_\star}{\Sigma^2} + \frac{m_\star^2 + \Sigma_\star^2}{2\Sigma^2}. \tag{61}$$

586  Therefore,

$$f(\theta) = \frac{1}{2}\log(2\Sigma^2) + \frac{\Sigma_\star^2}{2\Sigma^2} + \frac{(m - m_\star)^2}{2\Sigma^2} \tag{62}$$

587  In particular,

$$f(\theta) - f(\theta_\star) = \frac{1}{2}\log\left(\frac{\Sigma^2}{\Sigma_\star^2}\right) + \frac{\Sigma_\star^2 - \Sigma^2}{2\Sigma^2} + \frac{(m - m_\star)^2}{2\Sigma^2} \tag{63}$$

588  Note that, as expected, this corresponds to $D_A(\theta, \theta_\star)$, that we can also compute through Proposi-
589  tion F.1. We now write:

$$D_A(\theta, \theta^+) = -\frac{1}{2}\log\left(\frac{(\Sigma^2)^+}{\Sigma^2}\right) - \frac{\Sigma^2 - (\Sigma^2)^+}{2\Sigma^2} + \frac{(m - m^+)^2}{2\Sigma^2} \tag{64}$$

$$= -\frac{1}{2}\log\left((1-\eta)\left[1 + \eta\frac{(m-X)^2}{\Sigma^2}\right]\right) + \frac{(1-\eta)(\Sigma^2 + \eta(m-X)^2) - \Sigma^2}{2\Sigma^2} + \frac{\eta^2(m-X)^2}{2\Sigma^2} \tag{65}$$

$$= -\frac{1}{2}\log\left((1-\eta)\left[1 + \eta\frac{(m-X)^2}{\Sigma^2}\right]\right) - \frac{\eta}{2} + \eta(1-\eta)\frac{(m-X)^2}{2\Sigma^2} + \frac{\eta^2(m-X)^2}{2\Sigma^2} \tag{66}$$

$$= -\frac{1}{2}\log\left((1-\eta)\left[1 + \eta\frac{(m-X)^2}{\Sigma^2}\right]\right) - \frac{\eta}{2} + \eta\frac{(m-X)^2}{2\Sigma^2}. \tag{67}$$

590  Therefore,

$$f(\theta) - \frac{D_A(\theta, \theta^+)}{\eta} - f(\theta_\star) \tag{68}$$

$$= \frac{1}{2}\log\left(\frac{\Sigma^2}{\Sigma_\star^2}\right) + \frac{\Sigma_\star^2 - \Sigma^2}{2\Sigma^2} + \frac{(m - m_\star)^2}{2\Sigma^2} + \frac{1}{2\eta}\log\left((1-\eta)\left[1 + \eta\frac{(m-X)^2}{\Sigma^2}\right]\right) + \frac{1}{2} - \frac{(m-X)^2}{2\Sigma^2} \tag{69}$$

$$= \frac{1}{2}\log\left(\frac{\Sigma^2}{\Sigma_\star^2}\right) + \frac{1}{2\eta}\log\left((1-\eta)\left[1 + \eta\frac{(m-X)^2}{\Sigma^2}\right]\right) + \frac{\Sigma_\star^2}{2\Sigma^2} + \frac{(m - m_\star)^2}{2\Sigma^2} - \frac{(m-X)^2}{2\Sigma^2}. \tag{70}$$

591  Finally, $\mathbb{E}\left[(m - X)^2\right] = (m - m_\star)^2 + \Sigma_\star^2$, and so:

$$f_\eta(\theta) - f(\theta_\star) = \frac{1}{2}\log\left(\frac{\Sigma^2}{\Sigma_\star^2}\right) + \frac{1}{2\eta}\mathbb{E}\left[\log\left((1-\eta)\left[1 + \eta\frac{(m-X)^2}{\Sigma^2}\right]\right)\right], \tag{71}$$

592  which precisely corresponds to Equation (14). We now proceed to proving bounds on $\theta_\eta$ for $\eta < 1$.

 **F.2 Bounding the stochastic mirror descent variance $\sigma^2_{\star,\eta}$.**

594 Now that we have an explicit form for $f_\eta$, we can characterize its minimizer $\theta_\eta$, and use this to prove
595 results on $f_\eta(\theta_\eta)$, which will in turn lead to bounds on $\sigma^2_{\star,\eta}$. This is the core of Lemma 4.2.

596 *Proof.* Proof of Lemma 4.2. The proof will proceed in three different stages:

- Differentiating $f_\eta$ with respect to $m$ and $\Sigma^2$.

- Using these expressions to obtain bounds on the $(m_\eta, \Sigma^2_\eta)$ for which $\nabla f_\eta$ is 0.

- Plugging these bounds into the expression of $f_\eta$ to bound $\Sigma^2_\eta$.

600 **1 - Differentiating $f_\eta$.** Before differentiating, we rewrite:

$$f_\eta(\theta) - f(\theta_\star) = \frac{1}{2}\log\left(\Sigma^2\right) + \frac{1}{2\eta}\mathbb{E}\left[\log\left(1 + \eta\frac{(m-X)^2}{\Sigma^2}\right)\right] - \frac{1}{2}\log\left(\Sigma^2_\star\right) + \frac{1}{2\eta}\log(1-\eta)$$
(72)

$$= -\frac{1-\eta}{2\eta}\log\left(\Sigma^2\right) + \frac{1}{2\eta}\mathbb{E}\left[\log\left(\Sigma^2 + \eta(m-X)^2\right)\right] - \frac{1}{2}\log\left(\Sigma^2_\star\right) + \frac{1}{2\eta}\log(1-\eta).$$
(73)

601 Indeed, the two terms on the right are constant and so do not matter. If we differentiate in $m$, we
602 obtain:

$$\nabla_m f_\eta(\theta) = \mathbb{E}\left[\frac{1}{2\eta}2\eta\frac{m-X}{\Sigma^2}\frac{1}{\Sigma^2 + \eta(m-X)^2}\right] = \mathbb{E}\left[\frac{m-X}{\Sigma^2 + \eta(m-X)^2}\right].$$
(74)

603 Now, differentiating in $\Sigma^2$ yields:

$$\nabla_{\Sigma^2} f_\eta(\theta) = -\frac{1-\eta}{2\eta\Sigma^2} + \frac{1}{2\eta}\mathbb{E}\left[\frac{1}{\Sigma^2 + \eta(m-X)^2}\right] = \frac{1}{2\Sigma^2} - \mathbb{E}\left[\frac{(m-X)^2}{2\Sigma^2(\Sigma^2 + \eta(m-X)^2)}\right].$$
(75)

604 **2 - Obtaining bounds on $(m_\eta, \Sigma^2_\eta)$.** The solution to $\nabla_m f_\eta(\theta) = 0$ is $m = m_\star$. Indeed, it is direct
605 to verify that in this case, $\mathbb{E}\left[\frac{\tilde{X}}{\Sigma^2 + \eta\tilde{X}^2}\right] = 0$ since $\tilde{X} = m_\star - X$ is symmetric (with respect to 0). For
606 $m > m_\star$, $\mathbb{E}\left[\frac{\tilde{X}}{\Sigma^2 + \eta\tilde{X}^2}\right] > 0$ since we integrate the same values as the previous case, but now more
607 mass is put on the positive values (and similarly for $m < m_\star$). Note that this is the case regardless of
608 $\Sigma^2_\eta$.

609 We are now interested in $\Sigma^2_\eta$. Note that we will not get such a clean expression as for $m_\eta$, but only
610 bounds. From its expression, we deduce that $\nabla_{\Sigma^2} f_\eta(\theta_\eta) = 0$ can be reformulated as:

$$\mathbb{E}\left[\frac{(m_\eta - X)^2}{\Sigma^2_\eta + \eta_\eta(m-X)^2}\right] = 1$$
(76)

611 For the upper bound, we simply write that:

$$1 = \mathbb{E}\left[\frac{(m_\eta - X)^2}{\Sigma^2_\eta + \eta_\eta(m-X)^2}\right] \leq \mathbb{E}\left[\frac{(m_\eta - X)^2}{\Sigma^2_\eta}\right] = \frac{\Sigma^2_\star}{\Sigma^2_\eta},$$
(77)

612 from which we deduce that $\Sigma^2_\eta \leq \Sigma^2_\star$. Let us now introduce some $\alpha > 0$. We have that:

$$\mathbb{E}\left[\frac{(m_\eta - X)^2}{\Sigma^2_\eta + \eta(m_\eta - X)^2}\right] = \mathbb{E}\left[\frac{(m_\eta - X)^2}{\alpha - \alpha + \Sigma^2_\eta + \eta(m_\eta - X)^2}\right] = \mathbb{E}\left[\frac{(m_\eta - X)^2}{\alpha}\frac{1}{1 - 1 + \frac{\Sigma^2_\eta + \eta(m_\eta - X)^2}{\alpha}}\right]$$
(78)

613 We now use that for $u \geq -1$, $\frac{1}{1+u} \geq 1 - u$, and so:

$$\mathbb{E}\left[\frac{(m_\eta - X)^2}{\Sigma^2_\eta + \eta(m_\eta - X)^2}\right] \geq \mathbb{E}\left[\frac{(m_\eta - X)^2}{\alpha}\left(1 - \left[-1 + \frac{\Sigma^2_\eta + \eta(m_\eta - X)^2}{\alpha}\right]\right)\right]$$
(79)

$$= \mathbb{E}\left[\frac{(m_\eta - X)^2}{\alpha}\left(2 - \frac{\Sigma^2_\eta}{\alpha}\right) - \eta\frac{(m_\eta - X)^4}{\alpha^2}\right].$$
(80)

Now, recall that $m_\eta = m_\star$, so $X - m_\eta \sim \mathcal{N}(0, \Sigma_\star)$, leading to

$$1 = \mathbb{E}\left[\frac{(m_\eta - X)^2}{\Sigma_\eta^2 + \eta(m_\eta - X)^2}\right] \geq \frac{\Sigma_\star^2}{\alpha}\left(2 - \frac{\Sigma_\eta^2}{\alpha}\right) - \eta\frac{3\Sigma_\star^4}{\alpha^2}. \tag{81}$$

Rearranging terms, we obtain:

$$\frac{\alpha^2}{\Sigma_\star^2} - 2\alpha \geq -\Sigma_\eta^2 - 3\Sigma_\star^2, \text{ so } \Sigma_\eta^2 \geq \frac{2\alpha\Sigma_\star^2 - \alpha^2}{\Sigma_\star^2} - 3\eta\Sigma_\star^2. \tag{82}$$

We see that $\alpha = \Sigma_\star^2$ maximizes the right term, and we obtain the desired result, *i.e.*:

$$\Sigma_\eta^2 \geq (1 - 3\eta)\Sigma_\star^2. \tag{83}$$

Unfortunately, we see that this bound is only informative for $3\eta < 1$. For the rest of the cases, we will use the Markov inequality instead, which writes for all $a > 0$:

$$\mathbb{P}\left(\frac{(m_\eta - X)^2}{\Sigma_\eta^2 + \eta(m_\eta - X)^2} \geq a\right) \leq \frac{1}{a}\mathbb{E}\left[\frac{(m_\eta - X)^2}{\Sigma_\eta^2 + \eta(m_\eta - X)^2}\right] = \frac{1}{a}. \tag{84}$$

Yet,

$$\mathbb{P}\left(\frac{(m_\eta - X)^2}{\Sigma_\eta^2 + \eta(m_\eta - X)^2} \geq a\right) = \mathbb{P}\left(\frac{(m_\eta - X)^2}{\Sigma_\star^2} \geq \frac{a}{1 - \eta a}\frac{\Sigma_\eta^2}{\Sigma_\star^2}\right) = 2\mathbb{P}\left(\frac{X - m_\star}{\Sigma_\star} \geq \sqrt{\frac{a}{1 - \eta a}}\frac{\Sigma_\eta}{\Sigma_\star}\right). \tag{85}$$

Therefore, denoting $\Phi$ the cumulative distribution function of the standard Gaussian, we have:

$$2\left(1 - \Phi\left(\sqrt{\frac{a}{1 - \eta a}}\frac{\Sigma_\eta}{\Sigma_\star}\right)\right) \leq \frac{1}{a}, \tag{86}$$

and since $\Phi^{-1}$ is an increasing function, this leads to:

$$\sqrt{\frac{a}{1 - \eta a}}\frac{\Sigma_\eta}{\Sigma_\star} \geq \Phi^{-1}\left(1 - \frac{1}{2a}\right), \tag{87}$$

so that:

$$\Sigma_\eta \geq \sqrt{1 - \eta a}\frac{\Phi^{-1}\left(1 - \frac{1}{2a}\right)}{\sqrt{a}}\Sigma_\star \tag{88}$$

One can check that $\Phi^{-1}\left(1 - \frac{1}{2a}\right)/\sqrt{a} < 1$ for all $a$, which is consistent with the fact that $\Sigma_\eta^2 \leq \Sigma_\star^2$. Also note that for $\eta = 1$, a non-trivial bound would require $a < 1$, but then $\Phi^{-1}\left(1 - \frac{1}{2a}\right) \leq 0$ so (as expected), we cannot get better than $\Sigma_\eta^2 \geq 0$. However, the previous bounding (Equation (83)) is more precise for small $\eta$ since $\Phi^{-1}\left(1 - \frac{1}{2a}\right)/\sqrt{a} < 1 - c$ with $c > 0$ a constant regardless of $a$. In particular, for any $\varepsilon$, by using any $1 < a < 1/(1 - \varepsilon)$, we obtain that $\Sigma_\eta^2 \geq \alpha_\varepsilon \Sigma_\star^2$ for some constant $\alpha_\varepsilon$ that only depends on the $a$ that we choose. In particular, we can handle the cases $\eta = 1/2$ and $\eta = 1/3$ that gave trivial results $\Sigma_\eta^2 \geq 0$ with the previous bounds.

The last part consists in proving that $f_\eta(\theta_\eta) - f(\theta_\star) \geq \frac{1}{2}\log\left(\frac{\Sigma_\eta^2}{\Sigma_\star^2}\right)$. To do so, we start back from

$$f_\eta(\theta) - f(\theta_\star) = \frac{1}{2}\log\left(\frac{\Sigma^2}{\Sigma_\star^2}\right) + \frac{1}{2\eta}\mathbb{E}\left[\log\left((1 - \eta)\left[1 + \eta\frac{(m - X)^2}{\Sigma^2}\right]\right)\right],$$

and show that $\mathbb{E}\left[\log\left((1 - \eta)\left[1 + \eta\frac{(m_\eta - X)^2}{\Sigma_\eta^2}\right]\right)\right] \geq 0$. We start by the inequality $\log(1 + x) \geq \frac{x}{1 + x}$, leading to:

$$\mathbb{E}\left[\log\left((1 - \eta)\left[1 + \eta\frac{(m_\eta - X)^2}{\Sigma_\eta^2}\right]\right)\right] \geq \mathbb{E}\left[\frac{(1 - \eta)\left[1 + \eta\frac{(m_\eta - X)^2}{\Sigma_\eta^2}\right] - 1}{(1 - \eta)\left[1 + \eta\frac{(m_\eta - X)^2}{\Sigma_\eta^2}\right]}\right] \tag{89}$$

$$= \mathbb{E}\left[\frac{\eta(1 - \eta)\frac{(m_\eta - X)^2}{\Sigma_\eta^2} - \eta}{(1 - \eta)\left[1 + \eta\frac{(m_\eta - X)^2}{\Sigma_\eta^2}\right]}\right] \tag{90}$$

$$= \eta\mathbb{E}\left[\frac{(1 - \eta)(m_\eta - X)^2 - \Sigma_\eta^2}{(1 - \eta)\left[\Sigma_\eta^2 + \eta(m_\eta - X)^2\right]}\right] \tag{91}$$

Recall that the optimality conditions for $(m_\eta, \Sigma_\eta^2)$ write:

$$1 = \mathbb{E}\left[\frac{(m_\eta - X)^2}{\Sigma_\eta^2 + \eta(m_\eta - X)^2}\right] = \frac{1}{\eta}\mathbb{E}\left[1 - \frac{\Sigma_\eta^2}{\Sigma_\eta^2 + \eta(m_\eta - X)^2}\right], \tag{92}$$

so that

$$\mathbb{E}\left[\frac{\Sigma_\eta^2}{\Sigma_\eta^2 + \eta(m_\eta - X)^2}\right] = 1 - \eta. \tag{93}$$

Combining these, we obtain that

$$\mathbb{E}\left[\log\left((1-\eta)\left[1 + \eta\frac{(m_\eta - X)^2}{\Sigma_\eta^2}\right]\right)\right] \geq \eta\mathbb{E}\left[\frac{(1-\eta)(m_\eta - X)^2 - \Sigma_\eta^2}{(1-\eta)\left[\Sigma_\eta^2 + \eta(m_\eta - X)^2\right]}\right]$$

$$= \eta\left(\mathbb{E}\left[\frac{(m_\eta - X)^2}{\Sigma_\eta^2 + \eta(m_\eta - X)^2}\right] - \frac{1}{1-\eta}\mathbb{E}\left[\frac{\Sigma_\eta^2}{\Sigma_\eta^2 + \eta(m_\eta - X)^2}\right]\right) = 0,$$

which is the desired result.

The final result is obtained by plugging the lower bounds for $\Sigma_\eta^2$ into this bound, leading to either $\sigma_{\star,\eta}^2 \leq -\frac{1}{2\eta}\log(1 - 3\eta)$ for $\eta < 1/3$ or $\sigma_{\star,\eta}^2 \leq -\frac{1}{2\eta}\log\alpha_\varepsilon$ for $\eta < 1 - \varepsilon$.

$\square$

## F.3 Unrolling the recursions to derive actual convergence results.

### F.3.1 Proof of Theorem 4.3

Now that we have bounded the stochastic mirror descent variance $\sigma_{\star,\eta}^2$ in this setting, we can plug it into Theorem 3.1 to obtain finite-time convergence guarantees on the MAP and MLE estimators.

*Proof of Theorem 4.3.* Starting from Theorem 3.1, we obtain:

$$D_A(\theta_\star, \theta^{(k+1)}) \leq (1-\eta)D_A(\theta_\star, \theta^{(k)}) - \frac{\eta}{2}\log(1 - 3\eta) \leq (1-\eta)D_A(\theta_\star, \theta^{(k)}) + \frac{3\eta^2}{2}, \tag{94}$$

where the right term is replaced by $c_\varepsilon$ (where $c_\epsilon = -\frac{1}{2}\log\alpha_\varepsilon$) for $k \leq 3$. Taking $\eta = 1/k$ for $k > 1$ and multiplying by $k$ leads for $k > 3$ to:

$$kD_A(\theta_\star, \theta^{(k+1)}) \leq (k-1)D_A(\theta_\star, \theta^{(k)}) + \frac{3}{2k}. \tag{95}$$

Therefore, a telescopic sum leads to, for $n_0 > 0$:

$$(n + n_0)D_A(\theta_\star, \theta^{(n)}) \leq n_0 D_A(\theta_\star, \theta^{(0)}) + \frac{3}{2}\sum_{k=n_0}^{n+n_0}\frac{1}{k} + 2c_{1/2}, \tag{96}$$

and so, since $\sum_{k=n_0}^{n}\frac{1}{k} \leq \log(n + n_0 + 1) - \log(n_0)$:

$$D_A(\theta_\star, \theta^{(n)}) \leq \frac{n_0 D_A(\theta_\star, \theta^{(0)}) + (3/2)\log(1 + (n+1)/n_0) + \Gamma}{n + n_0}, \tag{97}$$

where $\Gamma = 2c_{1/2}$ and we actually have $\Gamma = 0$ for $n_0 > 3$.

$\square$

### F.3.2 $O(1/n)$ convergence result.

We now consider a different estimator (from the MAP and the MLE), which we construct in the following way:

- Choose $n_0 \geq 6$ and initial parameter $\tilde{\theta}^{(n_0)}$.
- Obtain $\tilde{\theta}^{(n)}$ by performing $n - n_0$ stochastic mirror descent steps from $\tilde{\theta}^{(n_0)}$ with step-sizes $\eta_k = 2/(k+1)$ for $k \in \{n_0, ..., n\}$.

This estimator is a modified version of the MAP, where $n_0$ controls how much weight we would like to put on the prior, and $\tilde{\theta}^{(n_0)}$ would typically be the same starting parameter as for the MAP estimator. This estimator is built so that we can use the convergence analysis from Lacoste-Julien et al. [16] and obtain a $O(1/n)$ convergence rate. Note that we make the $n_0 \geq 6$ restriction for simplicity to ensure that $\sigma^2_{\star,\eta} \leq 3/2$, but the result can be easily adapted to $n_0 \geq 2$.

**Proposition F.3.** *After $n - n_0$ steps, this modified estimator $\tilde{\theta}^{(n)}$ verifies:*

$$\mathbb{E} D_h(\theta_\star, \tilde{\theta}^{(n)}) \leq \frac{2n_0(n_0 - 1)}{n(n - 1)} D_h(\theta_\star, \tilde{\theta}^{(n_0)}) + \frac{6}{n}. \tag{98}$$

*Proof.* Let us note $D_k = \mathbb{E}\left[ D_h(\theta_\star, \tilde{\theta}^{(k)}) \right]$. In this case, using that $\sigma^2_{\star,\eta} \leq 3/2$, Theorem 3.1 writes (since $\mu = 1$):

$$D_{k+1} \leq (1 - \eta_k) D_k + \frac{3\eta_k^2}{2}. \tag{99}$$

At this point, we can multiply by $k(k + 1)$ on both sides, and take $\eta_k = \frac{2}{k+1}$ for $k \geq n_0$. Remarking that $1 - \eta_k = 1 - \frac{2}{k+1} = \frac{k-1}{k+1}$, we obtain that:

$$(k + 1)k D_{k+1} \leq k(k - 1) D_k + \frac{6k}{k + 1} \leq k(k - 1) D_k + 6. \tag{100}$$

Unrolling this recursion from $k = n_0$ to $k = n - 1$ (since $(k + 1)k D_{k+1} = L_{k+1}$, where $L_k = k(k - 1) D_k$), we obtain:

$$n(n - 1) D_n \leq n_0(n_0 - 1) D_{n_0} + \sum_{k=n_0}^{n-1} 6, \tag{101}$$

and the result follows by dividing by $n(n - 1)$, and using that $(n - n_0)/(n - 1) \leq 1$. $\qquad\square$

### F.4 The case of the MLE

For the MLE estimator, directly applying the mirror descent approach would require using $\eta_0 = 1$, starting from an arbitrary $\theta^{(0)}$ (that would not affect the results anyway). The problem in this case is that $D_h(\theta_\star, \theta^{(1)})$ is infinite since $\Sigma^{(2)} = 0$. This also means that we cannot start the stochastic mirror descent algorithm from $\theta^{(1)}$, since the recursion would still involve the infinite $D_h(\theta_\star, \theta^{(1)})$. Therefore, in the case of the MLE, considering that the first two samples are $X^{(1)}$ and $X^{(2)}$, then the first two points are:

$$m^{(1)} = X^{(1)}, \Sigma^{(1)} = 0 \text{ and } m^{(2)} = \frac{X^{(1)} + X^{(2)}}{2}, (\Sigma^{(2)})^2 = \frac{(X^{(1)} - X^{(2)})^2}{4}. \tag{102}$$

More generally, a direct recursion for the MLE leads to:

$$m^{(n)} = \frac{1}{n} \sum_{k=1}^{n} X^{(k)}, \quad (\Sigma^{(n)})^2 = \frac{1}{n} \sum_{k=1}^{n} (X^{(k)} - m^{(n)})^2. \tag{103}$$

From this, we derive that:

$$\mathbb{E}\left[ (\Sigma^{(n)})^2 \right] = \mathbb{E}\left[ (X^{(n)} - m^{(n)})^2 \right] \tag{104}$$

$$= \mathbb{E}\left[ \left( \left( 1 - \frac{1}{n} \right) X^{(n)} - \frac{n - 1}{n} m^{(n-1)} \right)^2 \right] \tag{105}$$

$$= \left( \frac{n - 1}{n} \right)^2 \mathbb{E}\left[ \left( X^{(n)} - m_\star - (m^{(n-1)} - m_\star) \right)^2 \right] \tag{106}$$

$$= \left( \frac{n - 1}{n} \right)^2 \mathbb{E}\left[ \left( X^{(n)} - m_\star \right)^2 + \left( m^{(n-1)} - m_\star \right)^2 \right] \tag{107}$$

$$= \left( \frac{n - 1}{n} \right)^2 \left( \Sigma_\star^2 + \frac{1}{n - 1} \Sigma_\star^2 \right) = \left( 1 - \frac{1}{n} \right) \Sigma_\star^2, \tag{108}$$

where (107) comes from the fact that $X^{(n)}$ and $m^{(n-1)}$ are independent with mean $m_\star$. Plugging this into the expression of $D_h(\theta_\star, \theta)$ for the MLE after $n$ steps, we obtain:

$$D_h(\theta_\star, \theta^{(n)}) = -\frac{1}{2}\mathbb{E}\left[\log\frac{(\Sigma^{(n)})^2}{\Sigma_\star^2}\right]. \tag{109}$$

Unfortunately, there is no closed-form for this expression for arbitrary $n$, hence the need for a more involved analysis, for instance through the mirror descent framework. For the case $n = 2$ however (which is the one we are interested in), we obtain that

$$D_h(\theta_\star, \theta^{(2)}) = -\frac{1}{2}\mathbb{E}\left[\log\left(\frac{X^{(1)} - X^{(2)}}{2\Sigma_\star}\right)^2\right] = -\frac{1}{2}\mathbb{E}\left[\log\frac{Y^2}{2}\right], \tag{110}$$

where $Y = \frac{X^{(1)} - X^{(2)}}{\sqrt{2}\Sigma_\star} \sim \mathcal{N}(0, 1)$. Therefore, this can simply be treated as a constant that we can precisely evaluate numerically (for instance remarking that $Y^2$ is gamma distributed and using results on logarithmic expectations of gamma distributions).

For $n > 2$, it is tempting to use the convexity of $-\log$ to use a similar reasoning, but this only leads to a constant bound on $D_h(\theta_\star, \theta^{(n)})$.

