# OpenReview forum: "Investigating Variance Definitions for Stochastic Mirror Descent with Relative Smoothness"
_NeurIPS.cc/2024/Conference — Submitted to NeurIPS 2024_

### Official Review · Reviewer_cHke · 2024-07-12

**Soundness:** 3
**Presentation:** 3
**Contribution:** 3
**Rating:** 5
**Confidence:** 4

**Summary:**

This paper investigates a new definition for the stochastic gradient variance in mirror descent.
Most existing analyses for stochastic mirror descent require a strongly convex distance generating function to bound the gradient variance.
This limits the their applications especially when this assumption fails.
In particular, Le Priol et al. (2021) have shown that the none of the existing convergence rates applies to Gaussian maximum likelihood.

This paper aims to fix this issue by proposing a new definition of gradient variance.
They show that the new definition is strictly stronger (more likely to hold in practice) than existing definitions, and derive convergence rates in convex setting.
The authors demonstrate an application of the new variance definition bounding the estimation error of MAP for one-dimensional Gaussian distributions.

**Strengths:**

- Analyzing stochastic mirror descent is hard when the distance generating function is not strongly convex.
This paper is a step towards generalizing mirror descent analyses.
In particular, I like Section 2.2 where the authors show that the proposed definition is strictly better than existing ones.

- The mirror descent analysis in this paper yields a non-asymptotic bound for the estimation error of Gaussian MAP. This seems to be a fundamental problem lacking theoretical guarantees based on Le Priol et al. (2021).
However, I am now knowledgeable enough to confirm the significance or novelty of this result in statistics.

**Weaknesses:**

While developing the new gradient variance definition is certainly interesting, I have the following concerns.

- The authors have shown that their gradient variance \\(\sigma_{\star, \eta}^2\\) is finite for every fixed step size \\(\eta\\).
The convergence rates in the convex setting are proved using constant step sizes, and thus the optimality gap does not vanish.
To make the optimality gap vanish, diminishing step sizes are often required, which is not covered in this paper.
Proving convergence with diminishing step sizes probably requires characterizing the average variance \\(\frac1T \sum_{t=1}^{T} \sigma_{\star, \eta_t}^2\\), which I think can be done only on a case-by-case manner depending on the specific application.

- The only case so far where this new definition shines while all other definitions fail is maximum likelihood estimation for one-dimensional Gaussian distributions.
This is very restrictive.
Is it possible to generalize this result to multivariate Gaussian distributions?
In addition, it would be great if the authors could provide other applications to further justify the necessity of this new definition.

Minor:
- Line 192: Add a period.
- Bad notation in Section 4.2: It might be confusing to use $\Sigma$ to denote the standard deviation.
Consider using a different letter like $s$ or $\tau$.

**Questions:**

See the questions in the weaknesses.

**Limitations:**

NA.

---

> ### Author Rebuttal · Authors · 2024-08-03
>
> Thank you for your review, we discuss your two main points below.
>
> **Diminishing step-sizes:** One would have to characterize some kind of average variance indeed. Yet, when $\sigma^2_\eta$ converges for small $\eta$ (Proposition 2.6), one can expect to obtain bounds of the form $\sigma^2_\eta \leq R^2$ for $\eta \leq \eta_0$, with $R^2_0 >0$ only dependent on $\eta_0$. This is for instance the case in our application. Therefore, while case-by-case bounds on the average variance are required, there are strong reasons to believe that these will generally hold.
>
> As a complement, diminishing step-size results are direct adaptations of the results in Section 3. The result in Section 4 actually relies on using diminishing step-sizes. In particular, the equivalent of Theorem 3.1 would write:
> \begin{equation}
>     \eta_t \left[ \mathbb{E} f_{\eta_t}(x^{(t)}) - f_{\eta_t}^*\right] +\mathbb{E} D_h(x_\star, x^{(t+1)}) \leq \prod_{k=0}^t(1 - \eta_k \mu) D_h(x_\star, x^{(0)}) + \sum_{k=0}^t \eta_k^2  \prod_{\ell=k}^t (1 - \eta_\ell \mu) \sigma_{\star, \eta_\ell}^2.
> \end{equation}
> The convex result (adaptation of Theorem 3.3) would write:
> \begin{equation}
>     \frac{1}{T+1}\sum_{k=0}^t \eta_k \mathbb{E}\left[f_{\eta_k}(x) - f_{\eta_k}^* + D_f(x_\star, x^{(k)})\right] \leq \frac{D_h(x_\star, x^{(0)})}{T+1} + \frac{1}{T+1}\sum_{k=0}^t \eta_k^2 \sigma^2_{\star, \eta_k}.
> \end{equation}
>
> **Generalization to multivariate Gaussian distributions:** Thank you for bringing out this point. We did not initially tackle the multivariate case since the unidimensional one already contained all the complexity of handling stochasticity with mirror descent. Yet, the multivariate can also be tackled with our approach by replacing quantities with their natural multidimensional equivalents (and in particualr $\log$ by $\log {\rm det}$).
>
> More precisely, the log-partition function writes up to constants:
> \begin{equation}
>     A(\theta) = - \frac{1}{2}\log{\rm det}(\Sigma^{-1}) + \frac{1}{2} m^\top \Sigma^{-1} m = - \frac{1}{2}\log\det(- 2\theta_2) - \frac{\theta_1^\top \theta_2^{-1} \theta_1}{4},
> \end{equation}
> where $\theta_1 = \Sigma^{-1} m$, and $\theta_2 = - \Sigma^{-1} /2$. From there, performing similar derivations, function $f_\eta$ writes:
> \begin{equation}
>     f_\eta(\theta) = \frac{d}{2\eta}(\eta + \log(1 - \eta)) + \frac{1}{2\eta} \mathbb{E} \log \det \left(I + \eta \Sigma^{-1} (m - x)(m - x)^\top\right) - \frac{1}{2}\log \det (\Sigma^{-1}),
> \end{equation}
> which is a direct transcription of what we had in the unidimensional case. From here, differentiating the above objective implies that the minimizer is $m_\eta = m_\star$ (as in the unidimensional case), and
> \begin{equation}
>     \frac{1}{\eta} \mathbb{E}\left[\left(\Sigma_\eta + \eta (m_\star - x)(m_\star - x)^\top\right)^{-1} \Sigma_\eta - I \right] + I = 0.
> \end{equation}
> Writing $X = \Sigma_\eta^{-1/2} (x - m_\star) \sim \mathcal{N}(0, \Sigma_\eta^{-1} \Sigma_\star)$, the previous expression writes:
> \begin{equation}
>     \mathbb{E}\left[ (I + \eta XX^\top)^{-1} XX^\top \right] = I,
> \end{equation}
> so in particular since $XX^\top$ is a rank-1 matrix, we can write:
> \begin{equation}
>     \mathbb{E}\left[ \frac{XX^\top}{1 + \eta \|X\|^2} \right] = I.
> \end{equation}
> This implies that the covariance of $X$ is isotropic, so $\Sigma_\eta^{-1} \Sigma_\star = \alpha I$ for some $\alpha > 0$.
>
> Taking the trace of this expression, we obtain
> \begin{equation}
>     d = \mathbb{E}\left[ \frac{\alpha Z}{1 + \eta \alpha Z}\right],
> \end{equation}
> where $Z$ is a chi-squared distribution with parameter $d$. Since $x \mapsto x / (1 + \eta x)$ is a concave function, we can apply Jensen inequality and obtain:
> \begin{equation}
>     d \leq \frac{\alpha d}{1 + \eta \alpha d}.
> \end{equation}
> In particular, this leads to
> \begin{equation}
>     \Sigma_\eta \geq (1 - \eta d) \Sigma_\star.
> \end{equation}
> Note that this is actually tighter than our 1d bound, which we can also improve in a similar way by just using Jensen on the same function in Equation (81) (instead of the other bound).
>
> Now that we have a bound on $\Sigma_\eta$, we can (similarly to the 1d case) substitute it into
> \begin{equation}
>     f_\eta(\theta_\eta) - f(\theta_\star) = \frac{1}{2} \log \det \left( \Sigma_\star^{-1} \Sigma_\eta\right) + \frac{1}{2\eta} \mathbb{E} \log \det \left((1 - \eta)\Sigma_\eta^{-1} \left[\Sigma_\eta + \eta (m_\eta - X)(m_\eta - X)^\top \right]\right).
> \end{equation}
> At this point, we use that $\log \det (A) \geq {\rm Tr}(I - A^{-1})$, so $\log \det (I + B) \geq Tr(I - (I + B)^{-1}) = Tr((I + B)^{-1} B)$, which is the multidimensional version of the $\log(1 + x) \geq x / (1 + x)$ that we use to obtain (89). Following similar derivations, we also obtain that the expectation of the log term is greater than 0, and we are left with:
> \begin{equation}
>     \sigma_{\star, \eta}^2 \leq - \frac{1}{2\eta} \log \det \left( \Sigma_\star^{-1} \Sigma_\eta\right) \geq - \frac{d}{\eta} \log (1 - \eta d).
> \end{equation}
>
> In the end, plugging everything, we obtain a $\tilde{O}\left(\frac{d^2}{n}\right)$ convergence result for the map with $n_0 > d$ independent samples (or actually anything that makes the prior full rank, and the condition $\eta \leq 1/d$). This is consistent with Appendix A.3 [17]. We write the result with a $\tilde{O}$ notation (that includes log factors) but they are non-asymptotic, similarly to the 1d case.
>
> In short, while the result are presented in the 1d case, no additional conceptual steps are required to tackle the d-dimensional one. We presented the main differences with the 1d proof in this rebuttal, and will make sure to add the full d-dimensional results to a revision of the paper.
>
> Another main application of our bounds are Poisson inverse problems: $D_{KL}(b, Ax)$, which are often solved using (stochastic) mirror descent and for which theoretical results are limited.
>
> We hope that we answered your question in a satisfying way, and are open to any further discussions.

---

> > ### Comment · Reviewer_cHke · 2024-08-09
> >
> > I acknowledge that I have read the rebuttal. I am satisfied provided that the authors add the extension to multivariate Gaussians.

---

### Official Review · Reviewer_6ts5 · 2024-07-13

**Soundness:** 2
**Presentation:** 3
**Contribution:** 2
**Rating:** 4
**Confidence:** 3

**Summary:**

This work revisits Stochastic Mirror Descent (SMD) proofs in the (relatively-strongly-) convex and relatively-smooth setting, and introduces a new (less restrictive) definition  of variance which can generally be bounded (globally) under mild regularity assumptions. Then this paper investigates this notion in more details, and show that it  naturally leads to strong convergence guarantees for stochastic mirror descent. Finally, this paper leverage this new analysis to obtain convergence guarantees for the Maximum Likelihood Estimator of a Gaussian with unknown mean and variance.

Problem:
In proof of Proposition 2, by the definition of $\sigma_{*,\eta}^2$, we can obtain that
$ \sigma_{*,\eta}^2 = \frac{\min_x f(x) - \min_x f_\eta(x)}{\eta}  $.
However,  $x_* =\argmin_x f(x)$ does not equal to $x_*' = \argmin_x f_{\eta}(x)$.
This will lead to $\sigma_{*,\eta}^2 \neq \frac{1}{\eta^2} D_h(x^*, x^+）$.

**Strengths:**

This work revisits Stochastic Mirror Descent (SMD) proofs in the (relatively-strongly-) convex and relatively-smooth setting, and introduces a new (less restrictive) definition  of variance which can generally be bounded (globally) under mild regularity assumptions. Then this paper investigates this notion in more details, and show that it  naturally leads to strong convergence guarantees for stochastic mirror descent. Finally, this paper leverage this new analysis to obtain convergence guarantees for the Maximum Likelihood Estimator of a Gaussian with unknown mean and variance.

**Weaknesses:**

No.

**Questions:**

In the proof of Proposition 2, by the definition of $\sigma_{*,\eta}^2$, we can obtain that
$ \sigma_{*,\eta}^2 = \frac{\min_x f(x) - \min_x f_\eta(x)}{\eta}  $.
However,  $x_* =\argmin_x f(x)$ does not equal to $x_*' = \argmin_x f_{\eta}(x)$.
This will lead to $\sigma_{*,\eta}^2 \neq \frac{1}{\eta^2} D_h(x^*, x^+）$.

---

> ### Author Rebuttal · Authors · 2024-08-03
>
> Thank you for your review.
>
> We agree with you that the minima are not the same, and this is actually a crucial point in the paper, otherwise we would directly obtain that $\sigma^2_\eta = \frac{1}{\eta}^2 \mathbb{E}D_h(x_\star, x_\star^+)$, which is not true. Yet, this holds asymptotically, as we show in Proposition 2.6.
>
> We are not sure which Proposition 2 you are referring to, but we never use the identity that you mention at the end of your summary in our proofs. In Proposition 2.2, we simply bound $f_\eta$ in terms of another function, but everything remains within the $\min$. We would be glad to detail any result you might be wondering about.
>
> We hope that you will reconsider your score in light of this answer, and would otherwise be happy to answer any other concern you might have.

---

### Official Review · Reviewer_H5hK · 2024-07-14

**Soundness:** 4
**Presentation:** 4
**Contribution:** 3
**Rating:** 6
**Confidence:** 4

**Summary:**

The paper proposes a new analysis of SMD using a newly introduced generalized variance notion. The benefit of the new analysis is demonstrated in the application to maximum a posteriori estimation of Gaussian parameters.

**Strengths:**

After introducing a new variance notion, the paper delves into comparison with other existing notions and shows that the proposed one is the largest meaningful notion. After a careful comparison, analysis of SMD is presented using this mild assumption. This analysis substantially departs from the results known in the literature. The demonstration of the use case of this new theory in the context of statistical estimation is also clear and adds more significance to the new theory.

**Weaknesses:**

Major:

As explained after theorem 4.3, the guarantees are derived for a reverse KL and may not imply anything on the desired quantity $f(\theta) - f(\theta_*)$. This of course, limits the contribution in this application significantly as non-asymptotic rates were known before.

Minor problems that I hope the authors can fix in the next revision.

1. Is the set C compact? If not, why the minimum exists in Proposition 2.2?

2. Cannot find where $x_*$ is defined. Why does it exist?

3. There is a small issue with indicies in equation (12) and in paragraph before. $\eta_{n} = \frac{1}{n_0+n+1}$, and the stochastic gradient should depend on the new sample $X_{n+1}$.

Update: meaningful results are obtained only for relatively strongly convex case (which is a stronger assumption than even strong convexity). In the convex case, a different (much stronger) definition is used. This becomes clear only after reading Appendix D. This limitation should be clarified in section 3.2, where convergence on some surrogate loss is shown. I will update my evalutation.

**Questions:**

Is it possible to consider other, non-Gaussian/non-conjugate, models?

---

> ### Author Rebuttal · Authors · 2024-08-05
>
> Thank you for your review and careful comments, we will make sure to fix all points you raised in the next revision.
>
> Before we start the point-by-point answer, we kindly ask you whether you could provide references for non-asymptotic rates on $D_f(\theta_\star, \theta)$ (since they probably do not exist for $f_\eta(\theta^{(n)})$, as we introduce this quantity). As a matter of fact, we do not only give non-asymptotic results on $D_f(\theta_\star, \theta^{(n)})$, but *anytime* results (for all $n$ such that it is finite). These results can then be converted on results for $D_f(\theta^{(n)}, \theta_\star)$ for $n$ large enough using similar arguments than in Le Priol et. al. (2021) (essentially, that Bregman divergences are close to quadratics when their arguments are close enough). Although we do not bound $f(\theta) - f(\theta_\star)$, we are not aware of equivalent results in the literature. However, we are not experts in Gaussian MAP estimation and would gladly compare to any reference you may provide.
>
> Please find the answers to your questions below:
>
> 1 - The set C is not necessarily compact. The minimum should thus be an infimum and can potentially be $-\infty$ at this point if the $f_\xi$ are not lower bounded (Corollary 2.3 ensures finiteness). Yet, the assumptions of Corollary 2.3 are not necessary to ensure finiteness.
>
> 2 - $x_\star$ is the solution to Problem (1), defined in Def 2 as the point such that $f(x_\star) = f^*$, where $f^*$ is the infimum of $f$, but we will define it more explicitly. We did not want to make its existence a blanket assumption since $\sigma_{\star, \eta}^2$ does not require $x_\star$ to exist, and neither do Theorems 3.1 and 3.3 (where the $D_h(x_\star, y)$ terms can be replaced by $\lim_{n\rightarrow \infty} D_h(u_n, y)$ for any sequence $u_n$ such that $f(u_n) \rightarrow f^\star$), since we do not use any property of $x_\star$ other than the fact that it's involved in defining $\sigma_{\star, \eta}$, which actually just involves $f^\star$. We used $x_\star$ here to facilitate reading.
>
> Yet, other works we compare to [7, 19] assume it exists, so we make this assumption to have comparable results in terms of interpretation. Similarly, when $x_\star$ exists, our variance definition has a nice interpretation in terms of ``gradients at optimum'' (Proposition 2.6). Similarly, assuming existence of $x_\star$ (and that it belongs to ${\rm int} C$) allows for a natural interpretation of $D_f(x_\star, x)$ in terms of 'gradient norm' (Proposition 3.4). To avoid any confusion, we will make existence of $x_\star$ a blanket assumption for convenience, and discuss case-by-case where it can be relaxed.
>
> 3 - Thank you for spotting this indexing issue.
>
> 4 - Regarding your update: We respectfully disagree with your statement on several aspects:
> - relative strong convexity is not stronger that strong convexity. For instance if $h(x) = -\log x$, $f(x) = h(x) + g(x)$ where $g(x) = a x$ for instance for some $a >0$ (but could also be another convex function), then $f$ is strongly convex relatively to $h$ but not strongly convex in the usual sense.
> - The fact that the control is not on $f(x^{(k)}) - f^\star$ is explicitly and clearly mentioned right after Theorem 3.3.
> - The results of Theorem 3.3 are meaningful, they are simply not function value results (see Proposition 3.4 and Proposition 2.5). It is an open question to know whether $f(x^{(k)}) - f(x_\star)$ can actually be bounded when using stochastic mirror descent without strong(er) assumptions on the variance (such as the ones we compare to in Section 2 or in Appendix D). The fact that we obtain convergence in terms of $f(x)$ in the deterministic case can be considered as an artifact of the deterministic setting, and we can recover it with our analysis: it corresponds to proving a result analog to Corollary 3.2 but for the convex setting, and in this case $E[f_\xi((x^{(k)})^+)] = f(x^{(k+1)})$. In the stochastic setting this simplification does not happen, and so we are 'stuck' with bounding either $f_\eta(x^{(k)})$ or $E[f_\xi((x^{(k)})^+)]$. Finally, note that Dragomir et. al. (2021) obtain even weaker results in the convex setting, since they do not bound $f_\eta(x^{(k)})$, and Hanzely and Richtarik (2021) use a definition that is comparably strong to the one discussed in Appendix D.
>
> In the end, we fully agree with you that convergence of function values directly would be appreciated. Unfortunately, it simply does not seem to be achievable in our setting.
>
> 5 - Is it possible to consider other non-Gaussian/non-conjugate models?
> Non-Gaussian models can be considered, as in this case Proposition 4.1 still applies for exponential families (as highlighted by Reviewer 35cP). However, several properties of the Gaussian distribution are used when bounding the variance, and so this work would have to be done again on a distribution-specific basis.
>
> For non-conjugate models, it would first be necessary to express the updates as resulting from stochastic mirror descent steps, which might be done in a case-by-case basis but maybe not in full generality.
>
>
> We hope that we have answered your concerns with the desired clarifications and that you will go back to your more positive evaluation. We are available for further discussion if this is not the case.

---

> > ### Comment · Reviewer_H5hK · 2024-08-11
> > **Asymptotic results and relative strong convexity**
> >
> > 1. It was a typo, I meant "asymptotic" results were already known before, e.g., discussed in Section 5 of Le Priol et al. [17]. My concern here is that whether the bound on the reverse KL is meaningful in this problem, because it seems to say nothing about $f(\theta) - f(\theta_*)$. The convergence in function value as in Hanzely and Richtárik [10] would be more suitable.
> >
> > 2. You're right, relative strong convexity is weaker than strong convexity only if h is strongly convex, which you don't assume here.
> >
> > 3. I am not entirely convinced about this statement "it simply does not seem to be achievable in our setting." Just because the proof doesn't go through easily in function value, does not mean the function value convergence is not achievable. Why the authors think it is not possible? If the authors can provide a convincing numerical experiment or a counter-example to show that the function value might not converge under these assumptions, I will be happy to increase my evaluation.
> >
> > 4. I believe the lack of convergence in function value in convex case is really important to clarify because it limits the applications of the results a lot. RE another application, Poisson inverse problems, suggested by the authors in the general response is also unclear if it is relatively strongly convex or not in general (even for a positive definite matrix).

---

> > > ### Author Response · Authors · 2024-08-13
> > >
> > > 1 - We agree that convergence in function values would be more suitable, unfortunately the assumptions from Hanzely and Richtarik [10] are too strong to be applied in this setting, which was one of the reasons for this work in the first place.
> > >
> > > 4 - The Poisson inverse problem is to the best of our knowledge not relatively strongly convex in general (the Hessian of the mirror can blow up if one $x_j$ goes to 0, whereas the Hessian of the objective remains bounded as long as $A_i^\top x$ remains bounded away from 0).
> > >
> > > Regarding our optimality criteria in the plain convex case, we would like to emphasize the following point (shared answer with that to Reviewer HwBA): our results are non-standard **with respect to standard Gradient Descent results**, not mirror descent ones. As a matter of fact:
> > > - Dragomir et. al. (2021) only control $D_h(x_\star, x)$ in the relatively strongly convex setting and $D_f(x_\star, x)$ in the convex one. We additionally control $f_\eta(x) - f_\eta^\star$ in both settings, under a weaker variance assumption.
> > > - Loizou et. al. (2021) do not consider a mirror descent setting (besides the "vanishing variance" issue, which we tackle below).
> > > - Hanzely and Richtarik (2021) make a very strong variance assumption, under which we have that $f_\eta(x) + \eta \sigma^2_{\rm sym} \geq \mathbb{E}f(x^+)$ (proof at the end of this comment), and so the control on $f_\eta$ gives a non-asymptotic control on $f$ in this case. Note that a similar control can be obtained from the variance Assumption from Appendix D.
> > >
> > > In particular, all other existing stochastic mirror descent analyses provide **weaker** results in terms of optimality metrics. We can recover control on function values either asymptotically or through stronger variance assumptions. To us, this is similar to the fact that standard relatively strongly convex Mirror Descent results only control $D_h(x_\star, x)$, but not $D_h(x, x_\star)$.
> > >
> > > Regarding the main application of the paper, we are in a similar position: as far as we know, existing results obtain bounds that hold either asymptotically or for $n$ large enough. This is because they bound other objectives, and then transfer their control to the right objective. Instead, we obtain anytime bounds, though on the reverse objective (and we are not aware of such bounds in other works).
> > >
> > > We have tried to obtain proper lower bounds to strengthen these claims on optimality criteria, but these are unfortunately very hard to come by as they would combine the difficulties of both the stochastic setting and the relative mirror descent setting (see, e.g, [1] for impossibility of acceleration in the mirror setting, which remained open for a significant time).
> > >
> > > We unfortunately do not have time to provide convincing numerical experiments in this rebuttal. One reason why we think that convergence in function values is not achievable without stronger assumptions is that our convex result is basically an equality (up to the removal of the $D_h(x_\star, x^{(k+1)})$ after the telescopic sum), so there is few room for improvement. Another is the one we gave you in the previous comment: the deterministic result follows the same scheme, but a simplification happens that allows to bound function values. This simplification does not happen in the stochastic case.

---

### Official Review · Reviewer_HwBA · 2024-07-17

**Soundness:** 3
**Presentation:** 2
**Contribution:** 2
**Rating:** 5
**Confidence:** 3

**Summary:**

This paper introduces a new variance assumption for the analysis of stochastic mirror descent (SMD) to handle cases where standard bounded variance assumption does not hold. The authors show this new assumption can be shown to hold under some regularity assumptions. The authors use the new results to show some convergence guarantees for MLE and MAP of a Gaussian with unknown mean and variance using the connection between this problem and SMD convergence guarantees.

**Strengths:**

The topic is definitely interesting and timely. Results for stochastic optimization without bounded variance assumptions are quite interesting. As shown in the prior literature, this task is especially subtle in the Bregman case. As the authors argue in detail, this difficulty is acknowledged in previous works such as [7] and [17]. It is neat that the authors show the importance of the new results by deriving convergence bounds for MAP/MLE of a Gaussian with unknown mean and variance by using the connection between these bounds and SMD in [17] (which itself is a nice connection). This adds a nice and clear motivation. The work makes some progress towards solving open questions from [17], while as the authors clearly explain, the open questions are still not completely solved.

**Weaknesses:**

I find the motivation of the paper and its application to MAP bounds interesting, however I have some concerns about writing and the strength of the derived results in the context of the application in Section 4. It seems necessary for the latter point to be clarified.

- Authors write after Theorem 4.3 that the open problem from [17] is not completely resolved because  the convergence is not shown for the desired quantity. In particular, the authors describe that the guarantee is for $D_A(\theta_*, \theta^{(n)})$ instead of $D_A(\theta^{(n)}, \theta_*) = f(\theta) - f(\theta_*)$. The authors then write that two quantities can be related asymptotically but they state: "but we might also be able to exploit this control over the course of the iterations". Can you make this point more precise? It is not clear to me what this last part is trying to describe. Is it meant to be understood as an open question or is it possible for the authors to derive the stronger result? Since the paper mentions at many places that showing convergence guarantees for MAP is an important contribution of the paper, it is important to justify the convergence metric used in the results for justifying the contribution of the paper fully.

- It might be better to replace MLE in the abstract to MAP since Section 4 is mostly about MAP.

- Abstract states a couple of times "strong convergence", I suggest to remove this since "strong convergence" has a precise meaning in infinite-dimensional optimization and usage in the abstract is confusing because of this. Clearly this is not how the authors are using this term, but it seems authors are using this as a subjective adjective, which is not necessary. By subjective, I mean that: how can one decide what convergence result is strong and what is not?

- Assumption 1 requires all $f_\xi$ are convex. This is rather strong since the standard assumption is $\mathbb{E} f_\xi$ to be convex. Can you discuss this more? According to Prop 4.1, this holds for the main application of the paper, but it might be worth discussing why componentwise convexity is needed.

**Questions:**

Please see the questions in the "Weaknesses" section.

- Can you describe more clearly "additional assumptions on the mirror, such as strong convexity (in the usual sense), to ensure bounded variance"?

- Line 179, 180: can you provide the proof for getting $\sigma_{\star, \eta}^2 \leq \mathbb{E} \| \nabla f_{\xi}(x_\star)\|^2$? I could not find this in Appendix B, can you let me know if I missed this?

- Corollary 2.3: Why does it follow that $\sigma_{\star, \eta}^2$ is finite? There is no lower bound on $\eta$, why can't one make this infinite for arbitrarily small $\eta$? The explanation in line 132-136 is also not clear. Can you provide clearer references and explanations for the claim you give in bold in this passage? To add to the confusion here, Line 206 mentions that the quantity that appears in this corollary can explode for $\eta \to 0$ by citing [19]. Can you first provide the precise pointer to [19] for this variance bound and then also explain the difference of this with Corollary 2.3?

- line 137: should change to "has already been investigated"

- Line 262-267: The authors say that the result is "non-standard" but they do not explain how meaningful it is, since the convergence is not shown for standard quantities. Can you please clarify?

- Bregman methods are used often with compact domains (especially simplex). It might be good to also mention this in the paper to say that some interesting applications may satisfy the bounded variance assumption but other interesting appllications do not. It might also help to provide more examples where we have unbounded domains and we use SMD.

- Section 4.2: Where did you define $R_{-}^*$?

**Limitations:**

The limitations are discussed clearly. The authors provided explanations after Theorem 3.3 and Theorem 4.3 to describe the limitations of their result.

---

> ### Author Rebuttal · Authors · 2024-08-03
>
> Thank your for your careful review and suggestions. We will change MLE to MAP in the abstract and remove the 'strong' adjective, which refered to the fact that we obtain similar guarantees for SMD as what we have in the `usual' setting for SGD, and was not specifically aimed at the statistical setting indeed.
>
> **Control over the course of iterations:** The sentence `we might be able to exploit this control over the course of iterations' refers to the fact that the convergence result in Section 4 completely discards the fact that we not only bound $D_A(\theta_\star, \theta^{(n)})$ but also $f_{\eta_n}(\theta^{(n)}) - f_{\eta_n}(\theta_{\eta_n})$. It might be possible to exploit the bound on $f_{\eta_n}(\theta^{(n)}) - f_{\eta_n}(\theta_{\eta_n})$ to obtain one on $D_A(\theta^{(n)}, \theta_\star)$, instead of wanting to obtain it from $D_A(\theta_\star, \theta^{(n)})$ for $\theta^{(n)} \approx \theta_\star$. Yet, we have not been able to leverage this control in a satisfying way, and so this sentence is more to be understood as an open question (and the fact that we control another metric related to the desired one).  We will make this clearer to avoid confusions.
>
> **Component-wise convexity:** This assumption is mostly used in Proposition 2.7 to use Bregman cocoercivity with the $f_\xi$ to compare with the results of Dragomir et. al., which require it (so we require it as well to compare to their definition). Our other results do not require convexity of $f_\xi$ directly, but we do use the fact that each $D_{f_\xi}$ is relatively smooth. We will discuss this more clearly.
>
> **Questions:**
> 1) This sentence refers to the variance expression of Line 191-192, which can be infinite in general, but is finite if $h$ is strongly convex (in the usual, not relative sense), since in this case the eigenvalues of $\nabla^2 h^*(u)$ are bounded uniformly in $u$.
>
> 2) The proof is not in Appendix B indeed, sorry for this. As sketched in the main text, we write that in this case, $\sigma^2_{\star, \eta} \leq \frac{f(x_\star) - f(x) + \frac{\eta}{2} \mathbb{E}\|\nabla f_\xi(x)\|^2}{\eta}$. Then, we use that $\frac{1}{2}\|\nabla f_\xi(x)\|^2 \leq \| \nabla f_\xi(x) - \nabla f_\xi(x_\star)\|^2 + \|\nabla f_\xi(x_\star)\|^2 \leq 2L D_{f_\xi}(x, x_\star) + \|\nabla f_\xi(x_\star)\|^2$, using cocoercivity of $f_\xi$ (which requires individual convexity of the $f_\xi$). Then, applying expectations, we have $\mathbb{E} \left[2\eta L D_{f_\xi}(x, x_\star)\right] = 2 \eta L D_f(x, x_\star) = 2\eta L \left[ f(x) - f(x_\star)\right]$, so that $\sigma^2_{\star, \eta} \leq \frac{(2\eta L - 1)}{\eta}\left[f(x) - f(x_\star)\right] + \mathbb{E}{\|\nabla f_\xi(x_\star)\|^2}$, leading to the desired result. This requires $\eta \leq 1/(2L)$, similarly to the Euclidean proof that uses this definition.
>
> 3) Finiteness is to be understood for a fixed $\eta$, which is already non-trivial due to the supremum on $x$. The precise reference in [19] is Assumption 2.1, and the difference is that this does not include the $1/\eta$ factor. As a result, the RHS of their Theorem 3.1 does not go to 0 as $\eta \rightarrow 0$. They study an adaptive step-size setting, but they would still get a non-vanishing RHS with constant step-size (making it go to 0) or vanishing step-sizes. Instead, $\eta \sigma^2_{\star, \eta}$ vanishes for $\eta \rightarrow 0$, ensuring that $D_h(x_\star, x^{(t)})$ can be arbitrarily close to 0 with enough steps and small enough step-sizes (which is not the case in [19]).
>
> 4) Clarifying non-standard: How meaningful this is very much depends on the applications. The guarantee on $f_\eta$ is meaningful in the sense that $f_\eta \rightarrow f$ for $\eta \rightarrow 0$ (Proposition 2.5). The result on $D_f(x_\star, x)$ can be seen on a control on the gradients of $f$, for instance through Proposition 3.4. Finally, $D_f(x_\star, x) \approx D_f(x, x_\star)$ if $x \approx x_\star$. Please note that existing results with relaxed variance assumptions (such as Dragomir et. al. (2021)), also only obtain convergence on  $D_f(x_\star, x)$ in the convex case. The 'non-standard' aspect is with respect to usual results on gradient descent.
>
> 5) Thank you for your suggestion, we will add a remark on this aspect in the paper.
>
> 6) $R^*_- =  \{ u \in R, u < 0 \} $.
>
> Thank you again for your many suggestions and comments, we hope that our results are clearer to you now and that you are willing to increase your score. We will gladly discuss any further question more in details.

---

> > ### Comment · Reviewer_HwBA · 2024-08-12
> >
> > Thank you for the explanations. Even though I see the potential value of the paper, I have still some questions and concerns.
> >
> > Question: You mention in the answer to me and Reviewer H5hK that the guarantee on $f_\eta$ is meaningful because $f_\eta \to f$. But I am not sure that this is an enough justification. This is especially because the paper wants to get non-asymptotic guarantees, whereas justifications such as $f_\eta \to f$ are only asymptotic. In particular, it seems you cannot convert nonasymptotic guarantee on $f_\eta$ to nonasymptotic guarantee on $f$, is this correct? If so, then I think one needs another way to justify that a nonasymptotic guarantee on $f_\eta$ is meaningful. I think a clearer connection is necessary here for different optimality metrics. Please let me know if I am missing something here.
> >
> > Concerns: The discussions where you are comparing variance definitions are quite confusing, at least to me. And this is perhaps one of the most important parts of the paper so the reader can understand the contributions. This was the case when I was writing the review and it is still the case even with the rebuttal. This needs to be really improved, in my opinion. For example, the discussion on lines 201-210 is quite confusing. You say "The vanishing variance term can be obtained by rescaling by $1/\eta$" which I am sure is very clear to the authors but not to the readers. Can you clearly describe what you mean by "vanishing variance term". It is confusing because earlier you talk about step size reducing the variance. But when you scale $f(x_*) - E[f_\xi(x_*^\xi)]$ by $1/\eta$, of course smaller $\eta$ makes this quantity larger. I guess what you call the "variance term" is different. But please be more explicit.
> >
> > Lastly, I want to emphasize that presentation is also extremely important since at the end we, as reviewers, and the authors want to make sure that the paper will be received well by our community. Especially given the volume of papers that appear in our conferences, I believe making papers as clear as possible and as readable as possible is extremely important to make sure that the community can read and understand the developments.
> >
> > If the authors can justify the non-standard guarantees (not on objective value) for the main application of the paper and promise to improve the presentation, I can push my recommendation to a "5" which is an accept rating, but my reasons are outlined above for why I am not comfortable raising the score more than that.

---

> > > ### Author Response · Authors · 2024-08-13
> > >
> > > **Regarding non-standard guarantees.** Your reasoning is correct and non-asymptotic guarantees on $f_\eta$ do not allow to obtain guarantees on $f$ a priori without further assumptions. Yet, we would like to emphasize that this non-standardness is **with respect to standard Gradient Descent results**, not mirror descent ones. As a matter of fact:
> > > - Dragomir et. al. (2021) only control $D_h(x_\star, x)$ in the relatively strongly convex setting and $D_f(x_\star, x)$ in the convex one. We additionally control $f_\eta(x) - f_\eta^\star$ in both settings, under a weaker variance assumption.
> > > - Loizou et. al. (2021) do not consider a mirror descent setting (besides the "vanishing variance" issue, which we tackle below).
> > > - Hanzely and Richtarik (2021) make a very strong variance assumption, under which we have that $f_\eta(x) + \eta \sigma^2_{\rm sym} \geq \mathbb{E}f(x^+)$ (proof at the end of this comment), and so the control on $f_\eta$ gives a non-asymptotic control on $f$ in this case. Note that a similar control can be obtained from the variance Assumption from Appendix D.
> > >
> > > In particular, all other existing stochastic mirror descent analyses provide **weaker** results in terms of optimality metrics. We can recover control on function values either asymptotically or through stronger variance assumptions. To us, this is similar to the fact that standard relatively strongly convex Mirror Descent results only control $D_h(x_\star, x)$, but not $D_h(x, x_\star)$.
> > >
> > > Regarding the main application of the paper, we are in a similar position: as far as we know, existing results obtain bounds that hold either asymptotically or for $n$ large enough. This is because they bound other objectives, and then transfer their control to the right objective. Instead, we obtain anytime bounds, though on the reverse objective (which no other people obtain).
> > >
> > > We have tried to obtain proper lower bounds to strengthen these claims on optimality criteria, but these are unfortunately very hard to come by as they would combine the difficulties of both the stochastic setting and the relative mirror descent setting (see, e.g, [1] for impossibility of acceleration in the mirror setting, which remained open for a significant time).
> > >
> > >
> > > **Comparison to other variance definition.** The "variance term" refers to the $\sigma^2$ term of the RHS of their Theorem 3.1, which is equal to $f(x_\star) - E\left[f_\xi(x_\star^\xi)\right]$, which does not go to 0 when $\eta \rightarrow 0$. This does not describe the actual behaviour of SGD, which is that the 'variance term' (as opposed to the term that depends on the initial conditions) vanishes when the step-size is small, allowing to obtain $f(x^{(t)}) - f(x_\star)$ (or other optimality criteria) arbitrarily close to 0 (with enough iterations and small enough step-size). We would obtain the same behaviour in our analysis if we were to use their variance definition.
> > >
> > > Instead, in our analysis, we replace $f(x_\star) - E\left[f_\xi(x_\star^\xi)\right]$ by $f(x_\star) - f_\eta^\star$, which goes to $0$ for $\eta \rightarrow 0$ (since the $1/\eta$ rescaled version has a finite limit as shown in Proposition 2.6). In this case, the optimality criterion can go arbitrarily close to 0 given enough iterations and small enough step-size.
> > >
> > >
> > > **Final comment.** We agree that presentation and clarity are extremely important, and thus thank you for helping us improve them for our paper. We hope these points are now clearer to you, and that you are now in a more comfortable position to recommend acceptance for our paper.
> > >
> > > **Proof of the claim.** We have that  $- \frac{1}{\eta} D_h(x, x^+) = \nabla f_\xi(x)^\top(x^+ - x) + \frac{1}{\eta} D_h(x^+, x) = \left(\nabla f_\xi(x) - \nabla f(x)\right)^\top (x^+ - x) + \frac{1}{\eta} D_h(x^+, x) + \nabla f(x)^\top (x^+ - x) $
> > > and so
> > >
> > > $- \frac{1}{\eta} D_h(x, x^+) = f(x^+) - f(x) + \left(\nabla f_\xi(x) - \nabla f(x)\right)^\top (x^+ - x) + \frac{1}{\eta} D_h(x^+, x) - D_f(x^+, x).$
> > >
> > > Taking expectations on both sides, we obtain using the relative smoothness of $f$ that:
> > > $f_\eta(x) \geq \mathbb{E} f(x^+)  - \eta \sigma^2_{\rm sym}$, leading to the desired result.
> > >
> > > [1] Dragomir, R. A., Taylor, A. B., d’Aspremont, A., & Bolte, J. (2022). Optimal complexity and certification of Bregman first-order methods. Mathematical Programming, 1-43.

---

### Official Review · Reviewer_35cP · 2024-07-22

**Soundness:** 2
**Presentation:** 1
**Contribution:** 3
**Rating:** 6
**Confidence:** 4

**Summary:**

This submission studies stochastic mirror descent (SMD) under quite mild conditions on the mirror map and objective function. More specifically, there are a variety of SMD analysis in the literature, but virtually all of them require strong conditions on the mirror map (such as strong convexity) that do not hold in cases where we only have relative smoothness (and/or relative strong convexity) of the objective function with respect to the mirror map. The authors propose a definition of variance of SMD that is better behaved under minimal assumptions. They show how this new variance can be used to obtain general convergence results for SMD. Finally, they show how the new variance definition for SMD can show some kind of non-asymptotic convergence rates for MLE and MAP of Gaussian parameter estimation with unknown mean and covariance, making partial progress on a conjecture posed by Le Priol et at.

**Strengths:**

This is an interesting paper that tackles a hard theoretical problem. I think it is of interest for researchers interested in mirror descent. The new definition of variance of SMD has interesting properties even under very mild assumptions, as the authors show when comparing the new definition with other definitions of SMD variance in the literature. Moreover, the results on Sec 4 already show how this is an interesting way to analyze SMD, and is likely to lead to follow-up work on the area.

So the strengths summarized in bullet points:
- Thorough comparison of new variance definition with other definitions in the literature and proof of finiteness under assumption 1.
- General convergence theorems of SMD under mild assumptions that recover known results in the deterministic case, showing this is may be a "natural" variance definition for SMD and useful for our understanding of SMD.
- Partial progress towards the conjecture of Le Priol et al.

**Weaknesses:**

In its current form, I have one main concern with the paper:
- Despite what is written at the beginning of the paper, **Assumption 1** is NOT a blanket assumption used throughout the paper. In fact, it appears only section 2 uses assumption 1. The rest of the paper uses a weaker assumption that is never clearly stated, which makes it hard to understand when the results hold or not.
This is likely to be a problem with presentation, but in its current form it is often not clear what are the assumption required at each point. Since the main point of the paper is to use a minimal number of assumptions, it is very important for those to be clearly stated.

A minor weakness is the lack of an example besides MAP/MLE. I could not easily think of a concrete example where I could apply the convergence results in sec 3 or 4. If the authors have an example besides MLE or MAP (even if a bit artificial), it would be great. For example, some example with a mirror map such as $- \log x$ would be interesting, but this is a minor suggestion, since it would be nice to see a concrete example of the use of the results in Sec 2 (the results in Sec 4 require a specialized bound on the variance)

Summary of weaknesses:
- Unclear requires assumptions for many of the results
- (Minor weakness) Lack of a concrete (even artificial) example of application of any of the theorems in Sec 3 beyond MAP/MLE (and the latter require specialized bounds on the variance).

**Questions:**

Currently I'm leaning towards accepting this paper (although borderline), but I'd expect the authors to address some of my concerns regarding assumptions (which, ultimately, are concerns about the correctness of the paper).
If the authors successfully address my concerns, I'll most likely increase my score. At this point I believe there should be mistakes with the assumptions, but my confidence if no high due to the lack of clarity regarding assumptions. I will be forced to decrease my score if another review or myself find a hard inconsistency in the proofs and the assumptions needed.

#### Questions about assumptions
- *$h$ should be of Legendre type?*: At some points the authors say they do not need assumption 1, but instead only need eq. (3) to hold for all iterates. One issue is that this is an assumption on the iterates, not on $h$, and leaves the conditions on $h$ way too open. In fact,  for many of the properties the authors seem to use in the appendix (Bregman divergence duality, for example), we usually require $h$ to **at least** be of Legendre type. Without this assumption it is unclear whether many of the derivations hold (I haven't checked the appendix carefully, but $h$ being of Legendre type is close to the minimal assumption for mirror descent to be well-defined). It is not clear that (3) implies a condition on $h$, and trying to prove something like this seems unecessary. The authors should either explain why they do not assume $h$ is of Legendre type (and show that all the properties they use actually hold even without assumption 1), or clearly state this assumption.
- *Formal statement of the assumption used?*: Ideally the authors would have the two different assumptions clearly stated, and then clearly mention either in each section or in the statement of each result which results depend on the assumptions. Also, eq. (3) seems to not be well-defined in general for general $h$ of Legendre type (for example, if $h = - \ln x$ the right-hand side might be negative if $x$ is large, regardless of $\eta$, which is a problem). Moreover, it would be interesting if the authors showed that the assumption on the iterates holds for a broad class of problems, such as general MAP for exponential families. This is because if $f_z(x) = h(x) - \langle x,z \rangle$ with $z \in \mathrm{cl}(\mathrm{dom} \nabla h)$ (here $z$ would be the sufficient statistics of one of the data points), then $\nabla h(x) - \eta \nabla f_z(h) = (1 - \eta)\nabla h(x) + \eta\nabla f_z(h)$, which is in the interior of the domain of $\nabla h$ for $\eta < 1$, showing that eq. (12) is indeed valid for exponential families. It is likely that the authors were aware of this already, but I believe clearly stating that this is true for general exponental families since Prop. 4.1 only covers the Gaussian case.
- *Assumption 1 is required for a result in sec 3, and this should be clearly mentioned*: Not a question, but related to the previous point: the authors say that the theorems in section 3 do not use assumption 1. While this seems to be true if read literally (not carefully checked), this might be read as all the *results* in section 3 not requiring Assumption 1, which is false: Proposition 3.4 uses Bregman co-coercivity, which in turn requires Assumption 1 to be true in general if I am not mistaken. The problem is that now the reader is left unsure what requires assumption 1 or not, putting into question whether the results in Sec 4 (which uses Sec 3) are correct or not.
- *Projected Gradient Descent*: The authors mention the 2-norm case a couple of times, but one should be careful regarding the set $C$ in this case. For $C \neq \mathbb{R}^d$  convex (say, the $\ell_2$-ball), assumption 1 fails to hold for $h = \frac{1}{2}\lVert\cdot \rVert_2^2$ since, in this case, the minimization problem may be attained at the boundary of $C$. This is an issue since (3) stops to be equivalent to (projected) gradient descent (either it becomes unprojected gradient descent or, if you add the indicator of $C$ to $h$, then $h$ may not be differentiable at the iterates $x$). In this case $h$ is not Legendre in $C$ (thus, having the assumption on $h$ would help prevent this case), and one might be able to handle this case by carefully handling the projections. I don't think it is interesting to try to do so, but this is a point that can bring even more doubts about the correctness of the paper.

#### Minor points
- 196-197: Not clear what "after the supremum has been taken" refers to exactly.
- 202-203: "Variance term" does not mean "variance", right? I think I understand what you meant on this lines, but this is a quite confusing passage and I had to re-read it a couple of times
- 232: The notation $\{0,T\}$ refers to a set with 2 elements, you did not mean that.
- 237: What are the assumptions of Lemma 4 of Dragomir? Maybe state it on the appendix, since it is very important for this result to not depend on Assumption 1, right? I'll check the lemma in detail later to verify.
- 278-283: This should definitely be mentioned earlier in the section, not at the end. Also, this should be formalized if possible.
- 318: Assumption 2 -> Definition 2 (Also, why assumptions/defs are numbered differently from theorems/propositions/lemmas?)
- 338: Not exactly clear what $\xi$  is here, use $X_n$ notation?
- This might be personal choice, but $\overline{x^{+}}$ is very big and stretches a lot of the lines you use this notation. $\overline{x}^+$  seems to be slightly better (although a bit cluttered). Don't feel like you need to change this if you disagree.

#### Post-discussion summary
Ultimately, this discussion on assumptions is more on the organization of the text and on the writing, not on the contributions of this submission. This is a very hard theoretical problem (which one can gauge by the amount of previous work with only partial results or restrictive assumptions). The partial solution to the conjecture on non-asymptotic convergence of MAP showcases the potential usefulness of these ideas for future work. Thus, I vouch for acceptance of this paper, since I believe it is of interest for the ML optimization community and can clearly be of inspiration for future work.

**Limitations:**

Although the authors are not explicit about some of the limitations of the results on sec 3, they do discuss how to interpret some of the results and limitations from their convergence rates.

---

> ### Author Rebuttal · Authors · 2024-08-03
>
> We would like to thank you for your detailed review, understand your concerns, and clarify these points below.
>
> **Questions 1 and 2**: From what we understand from your review, most concerns come from the `minimal assumptions on $h$' remark from lines 278-283. We realize that this remark can be confusing indeed as to the assumptions on $h$ that are necessary.
>
> Our intention was to emphasize the fact that Theorem 3.1 and Theorem 3.3 (and only these two results, as you point out) are mainly algebraic manipulations that explicit convergence in terms of relevant quantities. In particular, they only require assumptions on the iterates (in the form of Eq. (3)) because their goal is to express how quantities evolve from one iteration to the other. In fact, you can notice that the only inequality used in the proof of Theorem 3.1 is relative strong convexity. We also note that chaining (10) to obtain (6) requires $1 - \eta \mu \geq 0$, and so the condition $\eta \leq 1/\mu$, which we will make sure to add to Theorem 3.1 (we initially presented the theorems with the one-step versions (Eq 10) and then changed to (6)). Similarly, Theorem 3.3 requires $D_h(x_\star, y)$ to be positive for all $y$, and would otherwise include a $D_h(x_\star, x_{t})$ term.
>
> However, the fact that such minimal assumptions are required to write the derivations leading to Theorem 3.1 does not imply non-trivial results on $D_h(x_\star, x^{(t)})$. In particular *several quantities involved in Theorem 3.1 can potentially be infinite*. (6) still holds, but potentially with $f_\eta^\star = -\infty$ or $D_h(x_\star, x^{(0)}) = +\infty$. Finiteness can be obtained in different ways, such as using the general results given in Section 2, or using problem-specific bounds when these do not apply, as is done in Section 4.
>
> We understand that this can introduce confusion and we will highlight it very clearly. You are right that Assumption 1 is needed in general for some results in Section 3 (not the theorems though, but we agree that it can easily be read as `all results', which was not intended), which is why we present it as a blanket assumption.
>
> The `minimal assumption on $h$' was more intended to encourage readers to adapt inequalities such as (10) even for problems that do not satisfy the blanket assumption, if bounds on the various terms can be ensured in other ways. This is what we do for instance in Section 4 (in which just Eq. (10) is used, and $f_\eta^\star$ is bounded directly). We will rephrase it in that way and make sure to make this point extremely clear.
>
> Note that most of the Appendix is actually devoted to deriving results in Section 2 and Section 4. For Theorem 3.1. for instance we only prove Lemma C.1 with a very direct (and we believe verifiable) proof, which we encourage you to take a look at.
>
> **Question 3:** Thank you for your suggestion of writing Proposition 4.1 for exponential families more genrerally, we will make sure to add this to the paper.
>
> **Question 4**: Thank you for your remark, the L2 norm examples are more intended to give readers elements of comparison with more familiar expressions, we will add "unconstrained" explcitly in Line 175 since this is indeed the setting we had in mind. The (mirror) constrained case in which a projection term is added to (3) is discussed in Appendix E, though not tackled in full details.
>
>
> We thank you again for your feedback and are open to suggestions in order to better convey our message from the remark that the inequality holds under very mild assumptions, but results such as the ones derived in Section 2 are necessary to make the result meaningful in general.
>
> We hope we have lifted your doubts about correctness of the paper, and otherwise encourage you to take a look at the proof of Lemma C.1, or ask us to clarify any other concern you might have.

---

> > ### Comment · Reviewer_35cP · 2024-08-08
> > **Rephrasing questions 1 and 2**
> >
> > Thanks a lot for the authors for the detailed reply! I think the main point of my first two questions was not addressed, in part because I don't think I have properly written them. I will try to rephrase the questions, trying to be brief and clear. I do not expect a long answer either, but question 1 is something I am confused about and question 2 is asking for precise statement of the assumptions used.
> >
> > **Question 1: Do you need $h$ to be of Legendre type?** Assuming the mirror map is of Legendre type is a classical assumption to ensure everything works out in mirror descent. So it is not clear to me whether the authors need to go beyond Legendre type (which I do not think is the case) or if it is actually necessary for it to be of Legendre type. If so, it should be stated clearly the domain and everything (this would, for example, clear up the L2 norm case). I will take a look at appendix E later just in case.
> >
> > **Question 2: What are the formal statements of the assumptions used?**. Assumption 1 is a formal statement of the assumption used in Sec 3. But in Sec 4 the authors quickly mention that the assumption needs only to hold for the iterates... but then it is not clear what assumptions are needed on $h$. So there should be some sort of "Assumption 2" that clearly states all the assumptions on $h$ and/or the iterates. Then, at the beginning of each section the authors should explicitly say which assumption is being used (or mention on the theorems which assumption is used). Currently which were the assumptions necessary at each point was the part that confused me. Like I mentioned before, I think the authors should still explicitly assume that $h$ if Legendre (so that many of the properties of Bregman divergences work, for example) and formally state the assumptions mentioned in Sec 4.
> >
> > If I have the time I will also look at Lemma C.1. I just wanted to write a reply early in the discussion period to allow for the authors to write a reply without any rush. Again, I don't expect long answers, but these are my main points of confusion, that is why I am trying to rephrase the questions.

---

> > > ### Author Response · Authors · 2024-08-09
> > > **Answers to rephrased Questions 1 and 2.**
> > >
> > > Thank you for your quick reply, for engaging in the discussion, and for giving us time to respond. We greatly appreciate it.
> > >
> > > **Question 1.** We replace the Legendre-type assumption by Assumption 1, where twice continuous differentiability, strict convexity and well-definition of the the conjugacy problems (within the interior of $C$) ensure all the good properties we need for the derivations (since the gradient map is injective and so can be inverted on its image). We use this version of the Assumption following Dragomir et. al. (2021), to be able to write the 'Hessian formulation' of Bregman divergences, which we believe is often easier to interpret.
> > >
> > > If we do not want to assume that $h$ is twice continuously differentiable, then we would need to assume that it is Legendre on $C$ as you suggest, in order to be able to write duality for instance as you point out. We could also just assume Legendre and second-order differentiability when we would like to write out Hessians.
> > >
> > > Note that the L2 norm case is already cleanly handled with Assumption 1, since it imposes that the solution of the 'duality problem' is unique and lies within the interior of $C$ (which is essentially usually guaranteed by the Legendre assumption), which would not be the case if $h$ included a projection term.
> > >
> > > **Question 2.** We will remove the mention about relaxing assumptions right after Assumption 1, and replace it by the fact that Assumption 1 can be replaced by assuming that $h$ is Legendre if one does not have second-order continuous differentiability.
> > >
> > > Then, we will explicitly state in Section 2 that we give sufficient (but not necessary) conditions for several good properties of the variance to hold.
> > >
> > > Finally, we will give Assumption 2, a minimal assumption for Theorems 3.1 and 3.3 to hold, which are essentially well-definition of the iterates in the form of (3), and existence (in the sense of finiteness) of all the manipulated quantities. We will highlight that although these can be obtained using results in Section 2 (and the corresponding assumptions), they can also be obtained case-by-case for specific problems (as in Section 4).
> > >
> > > Thank you again for giving us the opportunity to clear up our answers, we are still available for further clarifications. We are fully aware that there is only so much time you can spend on a review, and we appreciate that you took the time to rephrase your questions so we can hopefully better address them. Regarding the appendix, we recommend first the proof of Lemma C.1, which is just a few lines without advanced arguments. Appendix E is longer and harder to read (introduces a few notations).

---

> > > > ### Comment · Reviewer_35cP · 2024-08-11
> > > > **Comment about assumptions and increase in score**
> > > >
> > > > I would like to thank the authors for carefully addressing my concerns.
> > > >
> > > > I would still suggest/insist the authors make the blanket assumption that $h$ is of Legendre type (note that $h$ does not need to be of Legendre type on $C$ but on maybe a larger set as long as you add the Bregman projection in the algorithm). The reason for that is two-fold:
> > > > - A mirror map of Legendre type is close to being a minimal set of assumptions on $h$ for mirror descent to be well-defined (which might even then not be enough, as discussed by Bubeck in "Introduction to Online Optimization", Sec. 5). I agree that Assumption 1 cleanly handles many of the examples covered in Sec 3, but when the authors try to change assumption 1 to only hold on the iterates, this makes it unclear what are the assumptions on $h$. I don't think assuming $h$ is Legendre will rule out any applications the authors consider and will make other researchers that worked on mirror descent like myself less worried about corner cases. For example, mirror maps of Legendre type allow the use of Bregman duality more cleanly (Lemma 5.1 of Bubeck in "Introduction to Online Optimization"). Without this assumption the gradient map stops being a bijection and one might be able to cook up weird corner cases;
> > > > - Mirror map being of Legendre type was used when relative smoothness was first introduced by Bauschke, Bolte, Teboulle, as well as in the by most classical texts covering mirror descent. For this work I can't see exactly the advantage of letting go of such a "classical" assumption. The disadvantage is that it makes readers acquainted with mirror descent worried about correctness of the paper. This is not a journal submission, so I am not verifying all the proofs, but from the rebuttals it seems like the authors essentially only look at functions of Legendre type, so you might as well say it explicitly to not make readers worry.
> > > > ---
> > > >
> > > > **Post-discussion summary:** Ultimately, this discussion on assumptions is more on the organization of the text and on the writing, not on the contributions of this submission. This is a very hard theoretical problem (which one can gauge by the amount of previous work with only partial results or restrictive assumptions). The partial solution to the conjecture on non-asymptotic convergence of MAP showcases the potential usefulness of these ideas for future work. Thus, I vouch for acceptance of this paper, since I believe it is of interest for the ML optimization community and can clearly be of inspiration for future work.

---

> > > > > ### Author Response · Authors · 2024-08-13
> > > > >
> > > > > We would like to thank you for the fruitful discussions and your positive post-discussion summary.
> > > > >
> > > > > We will make the blanket assumption that $h$ is Legendre as you suggest, as we agree with you that it cleans up many corner cases without strong drawbacks.

---

### Author Rebuttal · Authors · 2024-08-05

We thank all the reviewers for their careful evaluation and detailed feedback, which will greatly help us improve the clarity of our paper, in particular regarding the precise (and minimal) set of assumptions under which our results hold. We answer all questions point-by-point in the specific rebuttals, but first give application examples beyond the Gaussian MAP estimation.

As detailed in the introduction, mirror descent is hidden in many algorithms, and we believe that our framework would be valuable to analyze many more algorithms which do not benefit from non-asymptotic convergence guarantees yet. Making bridges between our analysis and existing results (hopefully improving on them) is an exciting though challenging task (as can be shown from the Gaussian MAP example).

To us, the other natural key application of ours results is Poisson inverse problems, which require solving problems of the form $\min_x D_{KL}(b, Ax)$. These problems can be analyzed in the same way as we do in Section 4. Yet, while our results give a precise framework for deriving convergence bounds for this other important problem, the end-to-end analysis is still challenging (as in the Gaussian MAP case), and out of scope of this paper.

On a side note, our results can improve on analyses using other variance definitions even when these are bounded. For instance, our variance definition is tighter than that of Dragomir et. al. (2021) even for strongly convex mirrors, since we only require to bound the smallest eigenvalue of the Hessian locally (around $x_\eta$) instead of globally (for strong convexity), so the constant might be much better.

Also note that, as discussed in the answer to Reviewer cHke, our results can be extended to handle multi-dimensional Gaussian MAP estimation.

We thank the reviewers again for their time in carefuly evaluating our paper, and are available for any further questions.

---

### Decision · Program_Chairs · 2024-09-25

**Decision:**

Reject

**Comment:**

This was a difficult decision. I think this is an incredibly important problem due to its implications about basic estimation ("statistics has no clothes"). So I was excited to see work on the topic but the result is ultimately not quite what we want. I do think the paper has some great insights and should be published, but the reviewers (a couple of which know the topic very well) felt that the paper was unclear on many points.

If this were a journal, I would be sending it back for a major revision. In the conference system, it is less clear what to do. I think the paper will benefit from a lot by a re-write, with the authors thinking carefully about what points the reviewers understood and including the additional examples brought up in the reviewer discussion. I ultimately fell on the side of recommending reject when considering that many other accepted papers have already gone through the process of making the contribution as clear as possible.

Nevertheless, I am looking forward to reading the revision.